# PRPO: Collaborative Online Policy Learning in Personalized RLHF

## Abstract

Personalizing Large Language Models (LLMs) requires capturing user preferences without centralizing private data, prompting a multi-agent local fine-tuning setup. While on-policy algorithms, as applied to RLHF, are well-suited for preference modeling, their use remains fundamentally single-agent. We present Peer-Referenced Policy Optimization (PRPO), an online policy-gradient method that lets privacy-constrained clients cooperate while keeping trajectories local. PRPO extends Proximal Policy Optimization (PPO) family and treats KL regularizer as a communication channel: each round, every client conditions its update on a composite reference policy created by peer-to-peer averaging of action distributions. This distribution-level exchange preserves trust-region stability and adds only modest overhead compatible with LoRA adapters. We provide theoretical support for PRPO through general observations and convergence guarantees under limited conditions. We evaluate PRPO on the set of ATARI and Minigrid challenges and in the standard RLHF summarization setting, where it surpasses local PPO — indicating that reference-policy sharing offers a practical path to scalable, privacy-preserving LLM personalization.

## 1 Introduction

Reinforcement Learning (RL) algorithms — both online methods such as PPO (Schulman et al., 2017b), (Ziegler et al., 2020) and GRPO (Shao et al., 2024), and offline approaches like DPO (Rafailov et al., 2024) — have proven highly effective in fine-tuning Large Language Models (LLMs) (Stiennon et al., 2022; Ouyang et al., 2022). These methods facilitate alignment with human preferences, safety objectives (Dai et al., 2023), and reasoning capabilities (DeepSeek-AI, 2025), leading to widespread use of LLM agents in a variety of downstream tasks.

Despite the success of general-purpose assistants like GPT-4 (OpenAI, 2024), the ability of LLMs to adapt to user-specific contexts — crucial for natural and effective human-AI interaction — remains an open challenge (Zhang et al., 2024b). Personalization raises a number of critical issues, including data privacy and fine-tuning efficiency (Li et al., 2024), which are especially pronounced in *personalized adaptation* settings (Liu et al., 2025). In these scenarios, each user is equipped with a lightweight, user-specific PEFT module (He et al., 2022) that captures their individual preferences.

A central problem in this context is how to enable collaborative learning among users: leveraging collective knowledge to improve model quality and learning efficiency, without incurring excessive communication overhead or violating privacy constraints (Liu et al., 2025). While a few recent works (Tan et al., 2024; Wagner et al., 2024; QI et al., 2024) have explored collaboration through LoRA (Hu et al., 2021) FedAvg-style (McMahan et al., 2023) weight aggregation schemes, these approaches operate outside the RL setting and focus on supervised fine-tuning for tasks such as language modeling and text classification.

**Contribution.** In this work, we propose an alternative direction that bridges the gap between personalized LLM methods and Reinforcement Learning with Human Feedback (RLHF). Specifically,

- we introduce **PRPO** (Peer-Referenced Policy Optimization), a cooperative extension to the Proximal Policy Optimization methods, which enables inter-agent communication during training;

- to facilitate this cooperation, we repurpose the KL penalty not only as a trust-region constraint, but also as a mechanism for coordination, conditioning each agent's policy on a composite reference that blends its own prior with aggregated peer policies;

- with this approach we introduce a new family of direct policy aggregation protocols that operate on action distributions rather than model weights. This enables flexible and well-structured communication design, ranging from fine-grained trust-aware gossip schemes to any operator that meaningfully acts on the policy space;

- we show how the PRPO method instantiates with three concrete proximal policy algorithms, yielding **PR-TRPO**, **PR-PPO**, and **PR-GRPO**, thereby demonstrating its applicability across TRPO (Schulman et al., 2017a), PPO (Schulman et al., 2017b), and GRPO (Shao et al., 2024);

- we provide theoretical motivation for PRPO through general observations proven under broad assumptions and strong convergence guarantees established under more restricted conditions;

- to validate our approach and ensure reproducibility, we develop a lightweight cooperative on-policy fine-tuning framework that would be available publicly;

- we evaluate PRPO as applied to PPO (PR-PPO) and GRPO (PR-GRPO) on RLHF text summarization tasks, as well as on classical reinforcement learning benchmarks such as MiniGrid (Chevalier-Boisvert et al., 2023) and ATARI (Bellemare et al., 2013), demonstrating the effectiveness of our approach for collaborative policy learning in both RLHF and pure RL domains.

## 2 RELATED WORK

Although this work is introduced through the lens of LLM personalization—narrowed down to *Personalized Adaptation* — the problem space spans multiple domains, including Federated Reinforcement Learning (FRL), Multi-Agent Reinforcement Learning (MARL), Federated LLM fine-tuning, Federated RLHF, and MARLHF. A complete survey would be required to outline the boundaries of this intersection. Here, we provide a brief overview of the most relevant areas that contextualize and motivate the proposed approach: a cooperative online policy gradient method designed for settings with limited local data, strong privacy requirements, and low communication overhead.

**Federated LLM Fine-Tuning.** Federated fine-tuning of LLMs has gained traction with the rise of frameworks such as OpenFedLLM (Ye et al., 2024) and Shepherd (Zhang et al., 2024a), which integrate PEFT techniques into classical Federated Learning algorithms like FedAvg (McMahan et al., 2023). These works demonstrate that LoRA-based updates are highly effective in reducing communication costs while maintaining strong privacy guarantees (Zhang et al., 2023). Novel techniques PEFT tailored for federated setting have also been introduced, such as FFA-LoRA (Sun et al., 2024). *These approaches validate the feasibility of distributed PEFT-based training but remain focused on supervised learning, leaving open the question of their applicability in personalization and RLHF.*

**Federated RLHF.** Some recent work explores Federated RLHF, primarily via preference modeling or offline RL. Approaches such as FedDPO (Ye et al., 2024), FedBis (Wu et al., 2025), and Plural-LLM (Srewa et al., 2025) reduce RLHF to supervised preference optimization—aggregating either local DPO-trained models or user-specific preference data into a centralized supervised training pipeline. However, *none of these works address **online** reinforcement learning in a federated setting, or support policy learning beyond model centralization.*

**Multi-Agent Reinforcement Learning (MARL).** Several classical MARL techniques offer relevant insights. IPPO (de Witt et al., 2020) shows the viability of independently trained local PPO agents, while MAPPO (Yu et al., 2022) leverages a centralized critic to coordinate agents. Most relevant is *CoPPO* (Wu et al., 2021), which *introduces step-size coordination* for communication among agents—achieving strong results on benchmarks like StarCraft II (Samvelyan et al., 2019). In contrast, *PR-PPO enables communication through reference policy aggregation*, which aligns more naturally with the RLHF training paradigm and the probabilistic interpretation of policies.

**Multi-Agent RLHF (MARLHF).** This area is still in its early stages, with only a few works focused on value-based RLHF, *preference modeling* in multi-agent games and *Nash equilibrium*

*learning* in general-sum environments (Zhang et al., 2025). As such, there *remains a significant gap in applying online policy-gradient methods* to cooperative RLHF scenarios.

**Personalized Adaptation.** Personalization in LLMs is a rich and evolving research area. As outlined in (Liu et al., 2025), personalization strategies span input-level (personalized prompting), objective-level (personalized alignment), and model-level *(personalized adaptation)*. This work focuses on the latter, *capturing the local characteristics with dedicated personalized model, rather than distinct objective design*, particularly in federated settings where clients maintain private PEFT modules. Prior studies (Tan et al., 2024; Wagner et al., 2024; QI et al., 2024) have explored supervised fine-tuning of personalized LoRA weights, but *none apply RLHF techniques*—despite their clear advantage for modeling user preferences.

Among them, the work of (Wagner et al., 2024) is especially relevant and worth a closer look. It addresses *"fine-tuning of large language models with limited local data availability"*, proposes *"decentralized LoRA training"*, and focuses on *"a peer-to-peer decentralized learning setting, where the existence of a central server is not assumed"*. It also *"explores different ways of building a trust-gossip aggregation graph across users"*. These aspects make it structurally and ideologically close to our setting.

However, key differences remain. First, (Wagner et al., 2024) restricts experiments to unsupervised language modeling, while our work addresses *Reinforcement Learning with Human Feedback* — a fundamentally different optimization regime. Second, (Wagner et al., 2024) aggregates LoRA weights, whereas we introduce a communication protocol that performs **policy (i.e., distributional) averaging**, more naturally aligned with the RLHF framework, particularly in the presence of KL constraints used in policy regularization.

## 3 METHOD

### 3.1 PROBLEM SETUP

Within the scope of *Personalized Adaptation*, we consider a setting involving $n$ agents, each with individual policy $\pi_i \in \Pi = \{\pi : \mathcal{S} \to \mathcal{P}(\mathcal{A})\}$, $i = 1, \ldots, n$, that operate on the common state space $\mathcal{S}$ and the action space $\mathcal{A}$ with transition probability distribution $P : \mathcal{S} \times \mathcal{A} \to \mathcal{P}(\mathcal{S})$, all pursuing the same objective (see Personalized Adaptation in Section 2): the discounted cumulative reward given the initial state $s_0$

$$f(s_0, \pi) = \mathbb{E}_{a_0, s_1, a_1, \ldots} \left[ \sum_{t=0}^{\infty} \gamma^t r(s_t) \right], \tag{1}$$

where $r : \mathcal{S} \to \mathbb{R}$ is the reward function, $a_t \sim \pi(\cdot \mid s_t)$ is the policy decision at time step $t$ and $s_{t+1} \sim P(\cdot \mid s_t, a_t)$ is the state drawn from the system dynamics based on the decision $a_t$ taken in the state $s_t$.

Although confined to the same system dynamics $P$ and aligned in goal $f$, the agents may differ significantly in their experience through distinct initial state distributions $\zeta_i^0 \in \mathcal{P}(\mathcal{S})$, and thus, in effect, aim different target

$$f_i(\pi) = \mathbb{E}_{s_0 \sim \zeta_i^0} f(s_0, \pi), \quad i = 1, \ldots, n. \tag{2}$$

Specifically, in the context of distributed LLM fine-tuning considered here — where the model's task is to predict the next token given a sequence of observed tokens — the action space is a finite vocabulary $\mathcal{V}$, and the state space is the set of all finite token sequences, denoted $\mathcal{V}^*$ (the Kleene star of $\mathcal{V}$). Agent policies $\pi_\theta(\cdot \mid \cdot)$ belong to a parameterized policy class $\Pi_\Theta = \{\pi_\theta : \mathcal{V}^* \to \mathcal{P}(\mathcal{V}) \mid \theta \in \Theta\}$, where each policy maps a token sequence to a distribution over next tokens. This class is defined by a shared LLM architecture with parameters $\theta$. The environment dynamics reduce to deterministic sequence extension: at time $t = 0, 1, \ldots$, given state $s_t \in \mathcal{V}^*$ and action $a_t \in \mathcal{V}$, the next state is the concatenated sequence $s_{t+1} = s_t a_t = s_0 a_0 a_1 \ldots a_t$.

This setting reflects our earlier assumption regarding the shared nature of the transition dynamics (1). Yet, heterogeneity is preserved through variation in the initial state distribution $\zeta_i^0$ across agents. This characterizes precisely the scenario of LLM personalization where each policy is trained on a unique local dataset $\{s_0\}_i = \mathcal{D}_i \sim \zeta_i^0$, that includes sensitive user information subject to privacy constraints.

However, this setup introduces several challenges. First, individual agents may lack sufficient local data to fine-tune models that accurately reflect user preferences. Second, privacy requirements prohibit direct aggregation or centralization of user data, necessitating that all training operations remain confined within local environments. Third, strictly local optimization raises further concerns regarding training efficiency and the ability to achieve high-quality models.

To address these challenges, prior work has explored parameter-efficient fine-tuning (PEFT), notably LoRA modules, as a solution. LoRA modules serve both as efficient local training mechanisms and compact, privacy-compliant units (Zhang et al., 2023) suitable for peer-to-peer communication among agents, as elaborated by Wagner et al. (2024).

From this prospective, the precise practical formulation of the problem is as follows:

> Given a shared, pre-trained policy frozen weights $\theta_0$, a set of trainable local LoRA adapters $\theta_1, \ldots, \theta_n$, local datasets $\mathcal{D}_1, \ldots, \mathcal{D}_n$ acquired from the corresponding distributions $\zeta_1^0, \ldots, \zeta_n^0$ and a common target $f(s_0, \pi)$ (1), that defines a set of personal objectives $f_i(\theta) = f_i(\pi_{\theta_0 + \theta})$ (2) — optimize each personalized policy $\pi_{\theta_0 + \theta_i}$ with respect to $f_i(\theta)$, through a local policy optimization procedure interleaved with peer-to-peer communication rounds for LoRA adapters exchange.

Previous work by Wagner et al. (2024) demonstrated the feasibility of this approach in an unsupervised next-token prediction task, leveraging weighted LoRA aggregation protocols and highlighted its superiority over local fine-tuning and classical federated averaging (McMahan et al., 2023).

In this work, we adopt this paradigm to the *online reinforcement learning* and propose a cooperative algorithm that employs an alternative communication protocol. We evaluate this method in native RLHF task (see Section 4.2) demonstrating its effectiveness in personalized adaptation, on par with validation in classical RL domains (see Section 4.1), where it outperforms both local training and existing federated baselines.

## 3.2 PEER-REFERENCED PPO

The PR-PPO is based on PPO (Schulman et al., 2017b) that updates policy through maximization of KL-penalized surrogate objective: [1]

$$\theta^{k+1} = \arg\max_{\theta} \widehat{\mathbb{E}}_t \left[ \frac{\pi_\theta(a_t|s_t)}{\pi_{\theta^k}(a_t|s_t)} \widehat{A}(s_t, a_t) - \beta \mathrm{KL}\left[ \pi_{\theta^k}(\cdot|s_t) \, \| \, \pi_\theta(\cdot|s_t) \right] \right], \quad (3)$$

where KL-penalty coefficient $\beta > 0$, $\widehat{A}(\cdot, \cdot)$ is an advantage estimator (e.g. GAE (Schulman et al., 2018)) and $\widehat{\mathbb{E}}_t$ denotes the empirical average over trajectory samples generated with old policy $\pi_{\theta^k}$.

The surrogate objective (3) constrains the model update to stay within a trust-region of the old policy $\pi_{\theta^k}$ as justified in TRPO to bound potential decrease in policy performance at each step (Schulman et al., 2017a). A common practice in RLHF further leverages the KL penalty as a mechanism to maintain a stable anchor — typically the pretrained SFT policy — throughout the optimization process, with sparse or no updates to the reference policy under the KL term (Ouyang et al., 2022).

In this work, given a number of policies to optimize $\{\pi_{\theta_i}\}_{i=1}^n \subset \Pi_\Theta$ (see Section 3.1), we go one step forward and condition a policy on its peers' aggregated reference to employ community value

$$\theta_i^{k+1} = \arg\max_{\theta} \widehat{\mathbb{E}}_t \left[ \frac{\pi_\theta(a_t|s_t)}{\pi_{\theta_i^k}(a_t|s_t)} \widehat{A}(s_t, a_t) - \beta \mathrm{KL}\left[ C_i \pi_{\theta^k}(\cdot|s_t) \, \| \, \pi_\theta(\cdot|s_t) \right] \right], \quad i = 1, \ldots n, \quad (4)$$

where $C_i : \Pi^n \to \Pi$ is the $i$-th component of communication operator $C : \Pi^n \to \Pi^n$ that is applied to a vector of reference policies $\pi_{\theta^k} = (\pi_{\theta_1^k}, \ldots, \pi_{\theta_n^k})$.

The exact nature of communication operator $C : \Pi^n \to \Pi^n$ may be of different origin, whether it be a constant matrix $C = \{c_{ij}\}$, or a complex operator $C^k$, evolving through the learning process. We would delve deeper the motivation behind its nature as well as the construction of a feasible class of such operators further, alongside the discussion of PR-TRPO (see Section 3.4).

---

[1]Technical details (e.g., clipping) are omitted in the derivations for clarity.

### 3.3 PEER-REFERENCED GRPO

In turn, GRPO (Shao et al., 2024) derives PPO objective (3) but swaps the terms of KL-Penalty and sets it in the reverse direction, resembling the approach of MDPO (Tomar et al., 2021a) that takes inspiration from the mirror-decent update rule. Furthermore, as the name suggests, GRPO exploits *grouped sampling* for the expectation estimate, that implies sub-sampling a trajectory group

$$\theta^{k+1} = \arg\max_{\theta} \widehat{\mathbb{E}}_{s_0 \sim \zeta^0} \widehat{\mathbb{E}}_{t|s_0} \left[ \frac{\pi_\theta(a_t|s_t)}{\pi_{\theta^k}(a_t|s_t)} \widehat{A}_{s_0}(r_t) - \beta \mathrm{KL} \left[ \pi_\theta(\,\cdot\,|s_t) \,\|\, \pi_{\theta^k}(\,\cdot\,|s_t) \right] \right], \quad (5)$$

where the empirical average $\widehat{\mathbb{E}}_{t|s_0}$ reduces to averaging over a group of trajectories $G(s_0)$ generated with old policy $\pi_{\theta^k}$ starting from a shared initial state $s_0$.

This allows GRPO to adopt *group relative advantage* that estimates advantage through normalization of rewards over the group in order to obviate the computation burden of value approximation

$$\widehat{A}_{s_0}(r_t) := \widehat{A}_{s_0}(r(s_t)) = \frac{r(s_t) - \mathrm{mean}(r(s_t) : s_t \in G(s_0))}{\mathrm{std}(r(s_t) : s_t \in G(s_0))}. \quad (6)$$

We adapt GRPO to multi-agent setting, analogous to (4) and introduce the PR-GRPO update rule

$$\theta_i^{k+1} = \arg\max_{\theta} \widehat{\mathbb{E}}_{s_0 \sim \zeta_i^0} \widehat{\mathbb{E}}_{t|s_0} \left[ \frac{\pi_\theta(a_t|s_t)}{\pi_{\theta_i^k}(a_t|s_t)} \widehat{A}_{s_0}(r_t) - \beta \mathrm{KL} \left[ \pi_\theta(\,\cdot\,|s_t) \,\|\, C_i \pi_{\theta^k}(\,\cdot\,|s_t) \right] \right], \ i = 1, \ldots n. \quad (7)$$

### 3.4 PEER-REFERENCED TRPO

Although we do not experiment with TRPO (Schulman et al., 2017a) in practice, we find it important to introduce PR-TRPO as it gives a clear prospective, viable for discussion of the method. In contrast to PPO (Schulman et al., 2017b), TRPO formulates policy updates as a hard-constrained optimization problem:

$$\theta^{k+1} = \arg\max_{\theta} \quad \widehat{\mathbb{E}}_t \left[ \frac{\pi_\theta(a_t \mid s_t)}{\pi_{\theta^k}(a_t \mid s_t)} \widehat{A}_t \right], \quad (8a)$$

$$\text{subject to} \quad \widehat{\mathbb{E}}_t \left[ \mathrm{KL} \left[ \pi_{\theta^k}(\,\cdot\, \mid s_t) \,\|\, \pi_\theta(\,\cdot\, \mid s_t) \right] \right] \le \delta; \quad (8b)$$

which adapts naturally to our cooperative approach:

$$\theta_i^{k+1} = \arg\max_{\theta} \quad \widehat{\mathbb{E}}_t \left[ \frac{\pi_\theta(a_t \mid s_t)}{\pi_{\theta_i^k}(a_t \mid s_t)} \widehat{A}_t \right], \quad (9a)$$

$$\text{subject to} \quad \widehat{\mathbb{E}}_t \left[ \mathrm{KL} \left[ C_i \pi_{\theta^k}(\,\cdot\, \mid s_t) \,\|\, \pi_\theta(\,\cdot\, \mid s_t) \right] \right] \le \varepsilon, \ i = 1, \ldots, n. \quad (9b)$$

The procedure (8a) – (8b) is motivated by the theory behind that guarantees monotonic policy improvement through unconstrained policy iteration procedure with TV-penalty that is replaced with KL-divergence provided that $\|x - y\|_{\mathrm{TV}}^2 \le \mathrm{KL}(x \,\|\, y)$. Though KL-divergence is a good choice for computational tractability, since the exact same theory originally allows the use of total variation we find it appropriate to lead the following discussion in a metric space setting.

In a metric space of policies $(\Pi, \rho)$, a $\rho$-TRPO solves the problem (8a) subject to *variational bound*

$$\rho(\pi^k, \pi^{k+1}) \le \delta. \quad (10)$$

Whereas $\rho$-PR-TRPO constrains each actor (9a) with its' *local commuted variational bound*

$$\rho(C_i \pi^k, \pi_i^{k+1}) \le \varepsilon_i. \quad (11)$$

A natural question arises: may we attain the original bound (10) with control of *local commuted bound (11)* so as to preserve the TRPO trust-region while enabling communication? In this regard, we propose to consider *uniformly $\gamma$-non-expansive communicators* in ambient Banach space $(\Sigma, \|\cdot\|)$.[2]

---

[2]This is indeed the case for policies, since distributions lie in the ambient Banach space of signed measures $\Sigma$ with the total variation norm. We would develop a more rigorous discussion with proofs in Appendix B.

**Definition 1.** We would call an operator $C : \Sigma^n \to \Sigma^n$ in Banach space $(\Sigma, \|\cdot\|)$ a uniform $\gamma$-non-expansion ($\gamma \leq 1$) as long as for $i = 1, \ldots, n$ and for all $x, y \in \Sigma^n$

$$\|C_i x - C_i y\| \leq \gamma \|x - y\|_\infty,$$

where $\|x - y\|_\infty$ is a $\ell^\infty$ product norm induced on $\Sigma^n$. In other words, $C$ is $\gamma$-Lipschitz in $(\Sigma^n, \|\cdot\|_\infty)$.

Equipped with this operator family, we may now state the following

**Observation 1.** *Given a uniform $\gamma$-non-expansion $C : \Sigma^n \to \Sigma^n$ in a Banach space $(\Sigma, \|\cdot\|)$ and a sequence of vectors in $\Sigma^n$: $\pi^0, \pi^1, \pi^2, \ldots$, **the local variation bound** $\|\pi_i^k - \pi_i^{k+1}\| \leq \delta$ **would hold uniformly, given the local commuted variation bound** $\|C_i \pi^k - \pi_i^{k+1}\| \leq \varepsilon$ **is satisfied uniformly** at the level*

$$\varepsilon \leq \frac{\delta - \gamma \|\pi^{k-1} - \pi^k\|_\infty}{2}. \tag{12}$$

And this suggests a path toward identifying 'stable' communication operators capable of maintaining the TRPO trust-region.

## 3.5 Communicators

To preserve the TRPO trust region via Observation 1, we require communicators satisfying uniform non-expansiveness (Definition 1). Appendix B introduces a general construction. Here, we describe practical subclasses used in our experiments.

*Example* 1 (Right-stochastic matrices). A right-stochastic matrix $S \in \mathbb{R}^{n \times n}$, with $s_{ij} \geq 0$ and $\sum_j s_{ij} = 1$, defines an operator $\mathcal{S} : \Pi^n \to \Pi^n$ via $\mathcal{S}_i(\pi) := \sum_j s_{ij} \pi_j$. For all $x, y \in \Sigma^n$:

$$\|\mathcal{S}_i(x) - \mathcal{S}_i(y)\|_{\mathrm{TV}} \leq \sum_j s_{ij} \|x_j - y_j\|_{\mathrm{TV}} \leq \|x - y\|_{\mathrm{TV}},$$

showing that $\mathcal{S}$ is a uniform 1-non-expansion.

The following communicators follow the same construction pattern, using stochastic matrices that may update at each communication round.

*Example* 2 (Self-preferred mean communicator). Given a preference level $p \in [0, 1]$, the matrix $C(p)$ has $c_{ii} = p$ and $c_{ij} = \frac{1-p}{n-1}$ for $i \neq j$. This allows each agent to retain a fixed share of its own policy while averaging the rest, balancing self-reliance and peer influence.

*Example* 3 (Similarity-based communicator). At round $k$, define:

$$c_{ij}^k \propto \sum_{s \in \mathcal{S}_{\mathrm{ref}}} \langle \pi_i(\cdot \mid s), \pi_j(\cdot \mid s) \rangle,$$

where the dot product is taken for vectors of policy values on a shared reference trajectory subset $\mathcal{S}_{\mathrm{ref}}$ and the resulting matrix $C^k = (c_{ij}^k)$ is row-normalized to ensure stochasticity.

*Example* 4 (Reward-based communicator). At round $k$, set:

$$c_{ij}^k = \frac{r_j}{\sum_{k=1}^n r_k},$$

where $r_j$ is the average episodic return of agent $j$ since the last communication round. This communicator biases aggregation toward higher-performing agents.

These operators are evaluated in Sections 4.1 and 4.2.

## 4 Experiments

We evaluated the PRPO algorithms in two distinct domains: classical reinforcement learning environments and RLHF alignment. We began with relatively lightweight experiments in the ATARI and MiniGrid environments to validate the effectiveness of our method and compare it against standard federated learning baselines. We then conducted experiments in a more computationally intensive RLHF setup to evaluate PRPO variants against centralized algorithms and assess the impact of peer communication.

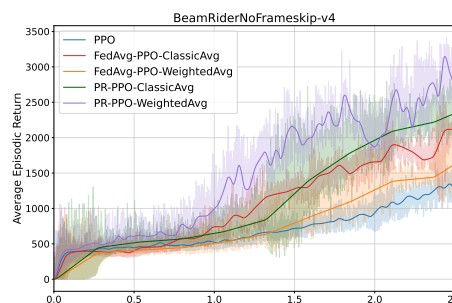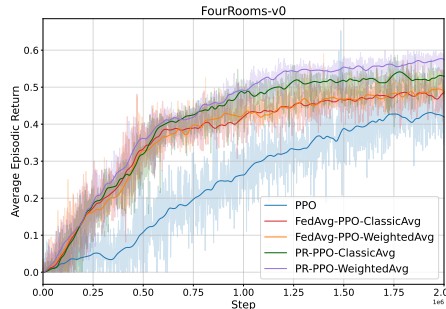

Figure 1: Performance of PR-PPO, isolated PPO and FedAvg-PPO on the `BeamRiderNoFrameskip-v4` (left) and `FourRooms-v0` (right). For PR-PPO and FedAvg-PPO the figure shows average performance within the group.

### 4.1 CLASSICAL RL

In this section, we consider a series of experiments in discrete multi-agent reinforcement learning environments, specifically focusing on Minigrid Chevalier-Boisvert et al. (2023) and Atari Bellemare et al. (2013). We investigate the adaptive choice of the communication matrix and consider more complex baselines. [3]

**Experimental Setup.** We consider `FourRooms-v0`, `DoorKey-6x6-v0`, and `DistShift2-v0` environments from Minigrid, and `BeamRiderNoFrameskip-v4` and `AsterixNoFrameskip-v4` environments from Atari. Our evaluation compares the performance of PR-PPO, isolated PPO Schulman et al. (2017b) and FedAvg McMahan et al. (2023)-PPO.

In all non-isolated experimental setups, we consider groups comprising three agents. Within these groups, agents communicate through a communication protocol. Partially similar to Wagner et al. (2024), we consider three different choices of the communication matrix (see the corresponding case in Section 3.5). The first is classical averaging, where each agent contributes equally with $c_{ij} = 1/N$. The second is adaptive reward-based averaging from Example 4. The third approach is also adaptive but policy-based averaging from Example 3.

The configurations for all algorithms were carefully fine-tuned through a grid search over a comprehensive set of hyperparameters. Each figure depicts the performance of the best configurations, averaged over ten random seeds. Additional details are provided in Appendix D.

**Results.** We use classical and policy-based averaging in Minigrid environments, and classical and reward-based averaging in Atari environments. The results are presented in Figure 1 and Appendix D.

From Figure 1, it is evident that our proposed PR-PPO outperforms both FedAvg-PPO and isolated PPO baselines in all settings. This means that our approach is better than not only local training but also the classical approach to federated learning. We also observe that both adaptive averaging techniques consistently achieve higher rewards compared to classic averaging, suggesting that dynamic weight-selection protocols might be favored in different practical scenarios.

### 4.2 RLHF

For our experiments with language models, we use the PR-PPO and PR-GRPO algorithms, as their centralized counterparts (PPO and GRPO) have proven effective and are widely used in the RLHF stage of language model alignment. Additional details are provided in Appendix D.

**Data.** In the RLHF setting, we focus on the summarization task, following early work in the field (Ziegler et al., 2020; Stiennon et al., 2022), as it is both computationally lightweight and widely used in research. We utilize the TL;DR dataset (Völske et al., 2017), which consists of Reddit posts

---

[3]The code of this experiment is publicly available in our FedRL repository.

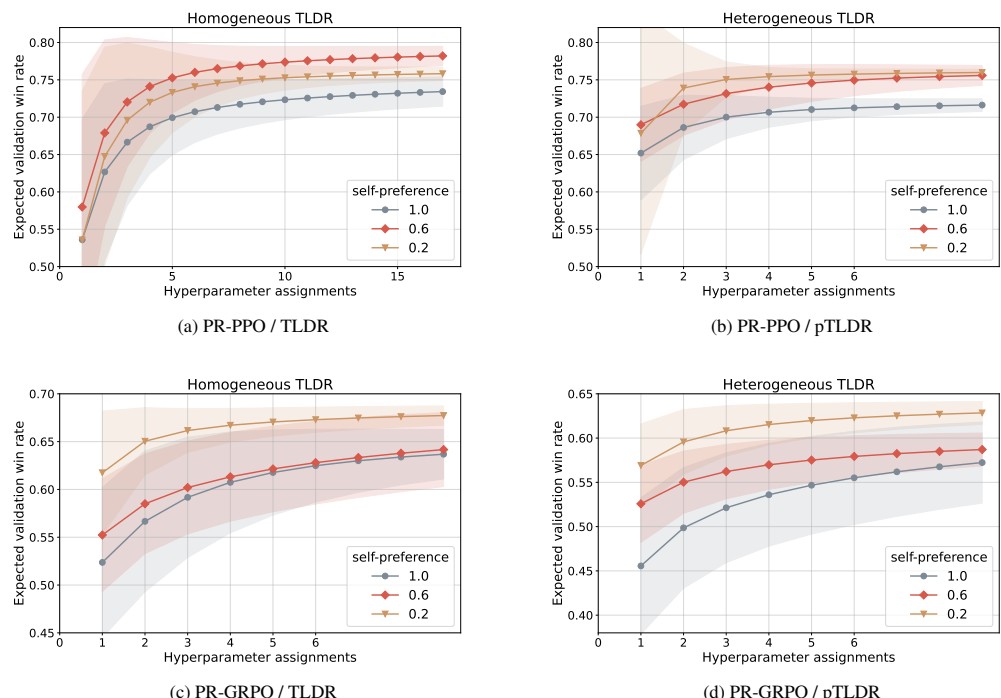

Figure 2: Expected validation win rate curves for PR-PPO and PR-GRPO in two data regimes at different levels of self-preference.

paired with user-written summaries. [4] We evaluate two setups for the TL;DR task: homogeneous and heterogeneous data distributions. In the homogeneous setup, agents receive identically distributed data (i.e., each agent owns a randomly sampled subset of the TL;DR dataset). In the heterogeneous setup, each agent is assigned a distinct, client-specific dataset. This is achieved by partitioning the TL;DR dataset into subsets corresponding to subreddits. We have prepared and published this thematic TL;DR dataset on our Hugging Face page.

**Implementation.**    We implemented the PR-PPO and PR-GRPO algorithms using the Torchtune framework (torchtune maintainers and contributors, 2024) as a base. The code is publicly available in our repositories. [5]  The training pipeline is distributed: each agent is assigned to a separate GPU to emulate a realistic federated setup. Agents are trained using LoRA adapters and, during communication rounds, emulate exchanging these adapters to construct locally aggregated reference policies. Communication follows a fixed protocol (see Example 2 and Example 4) and occurs at a predefined frequency. For SFT and reward training, we use the standard Hugging Face TRL library trainers (von Werra et al., 2020), following the guidelines from Huang et al. (2024).

**Evaluation.**    Evaluating alignment quality in language models is non-trivial. A widely adopted approach is the use of LLM-as-a-judge, explored in detail by Zheng et al. (2023). In our experiments, we also employ a large LLM to serve as a judge. Since the TL;DR dataset includes human-written reference summaries, we assess model outputs by comparing them against these references. Our primary metric is the win rate — the proportion of model-generated summaries judged to be better than the human-written ones on the validation set. Following Dodge et al. (2019; 2021), we report performance using the Expected Validation Performance (EVP) metric. EVP is computed by sampling multiple subsets of training runs, calculating the maximum validation score in each, and averaging

---

[4]We use the open-source trl-lib/tldr dataset for Supervised Fine-Tuning and PPO training, and trl-lib/tldr-preference for reward model training. The heterogeneous TL;DR dataset is available at anonymous-organization/tldr-thematic.

[5]See our implementation of PR-PPO and PR-GRPO in the TunePPO repository. Dataset preparation and reward model training are available in the PrePPO repository.

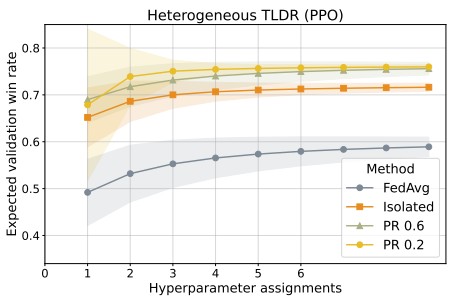 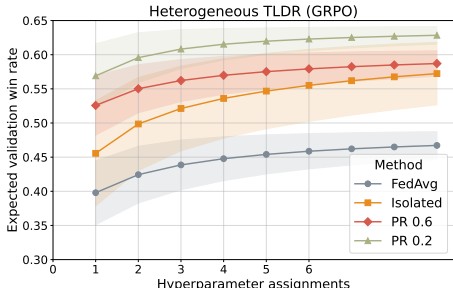

Figure 3: Expected validation win rate for isolated PPO and GRPO; FedAvg-PPO, FedAvg-GRPO; PR-PPO and PR-GRPO at different levels of self-preference.

these maxima to reflect performance under realistic compute budgets. In our case, the validation score is the win rate. This approach allows us to quantify how performance scales with the number of hyperparameter trials — or, equivalently, with the computational resources invested. EVP thus provides a more robust assessment of algorithm performance than reporting only the single best result.

**Experimental Setup.** We train seven Mistral-7B agents (Jiang et al., 2023) using the Self-Preferred Mean Aggregation communication protocol (see Example 2). We use Qwen3-32B (Team, 2025) as the LLM evaluator, as it is efficient enough to run on a single 80GB GPU while achieving performance comparable to the widely used LLaMA3.3-70B-Instruct model (Meta-AI, 2024). Additionally, it supports a reasoning mode, which is beneficial for evaluation tasks. All experiments were run on a cluster with eight NVIDIA H100 80GB GPUs: seven used for training and one reserved for evaluation.

**Results.** We compute EVP while sweeping three hyperparameters: the KL coefficient (Section 4), the self-preference in Self-Preferred Mean Aggregation (Example 2 in Appendix B.3) and the update interval (steps between reference-policy updates). EVP plots are grouped by self-preference; shaded regions show the standard deviation. From Figure 2, for both PR-PPO and PR-GRPO, communicating agents (self-preference < 1) consistently outperform isolated agents (self-preference = 1) in homogeneous and heterogeneous setups. Overall, in the RLHF setting, cooperation between language models yields substantial gains, underscoring the value of decentralized training and showing that even lightweight communication protocols can deliver measurable alignment benefits.

We also adapted Federated Averaging (FedAvg) to PPO and GRPO and compared it to our method and local fine-tuning. While FedAvg performs reasonably in simpler environments (e.g., Atari), it consistently underperforms in LLM fine-tuning (Figure 3), suggesting weight-level aggregation is too coarse for aligning complex policies and motivating more expressive policy-level sharing. Below we provide the full table summarizing experiments that reached a solid level of statistical reliability.

| Method | Data | Isolated | FedAvg | PR (Ours) | | Win Rate Gain | | | |
|--------|------|----------|--------|-----------|--------|----------------|--------|--------|--------|
| | | | | | | vs Isolated | | vs FedAvg | |
| | | (1.0) | (1/7) | 0.6 | 0.2 | 0.6 | 0.2 | 0.6 | 0.2 |
| PPO | TLDR | 0.734 | - | **0.782** | 0.758 | **+6.5%** | +3.3% | - | - |
| | PTLDR | 0.716 | 0.589 | 0.755 | **0.759** | +5.4% | **+6.0%** | +28.2% | **+28.9%** |
| GRPO | TLDR | 0.636 | - | 0.641 | **0.677** | +0.8% | **+6.4%** | - | - |
| | PTLDR | 0.572 | 0.467 | 0.587 | **0.628** | +2.6% | **+9.8%** | +25.7% | **+34.4%** |

Table 1: Expected validation win rate comparison for different approaches: isolated local fine-tuning, FedAvg and ours Peer-Referenced optimization under different self-preference levels (0.2 and 0.6). In two data regimes: homogeneous (TLDR) and heterogeneous (PTLDR).

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

# A  LIMITATIONS AND FUTURE WORK

While our work makes several conceptual and empirical contributions across domains, it also presents limitations stemming from the scope decisions necessary to keep the study tractable. First, we do not directly analyze the scalability and privacy-preservation guarantees of PRPO, and instead refer to prior work that coincides with our approach in subjects (e.g. LoRA exchange in general) unrelated to our direct contribution such as (Zhang et al., 2023; Wagner et al., 2024). Second, although our method allows for a range of communication strategies, in RLHF summarization domain we focus on a simplified setting where agents apply a *self-preferred mean aggregation* and explore *performance-based adaptive averaging* and *similarity-based adaptive averaging* in classical RL domains. Third, although we outline how PRPO generalizes to multiple proximal policy algorithms, our empirical evaluation is limited to PR-PPO and PR-GRPO, leaving systematic benchmarking of PR-TRPO to future work. Fourth, the theoretical results we present offer foundational insight and convergence intuition, but do not constitute a full theoretical framework; rigorous analysis under more general conditions for locally non-optimizable objectives requiring cooperation — remains an open problem situated within the broader field of multi-agent and multi-objective optimization. Finally, our experiments focus on relatively lightweight tasks, such as summarization, and use modest model sizes due to computational constraints. We leave validation on more demanding domains — including question answering, instruction tuning, and reasoning with larger models — to future work.

# B  ON COMMUNICATORS AND TRUST-REGION

In Sections 3.4 and 3.5, we briefly introduced the conservation of the $\rho$-TRPO trust-region (10) within the $\rho$-PR-TRPO optimization (9a)–(11), along with the role of communicators in maintaining this conservation in order. In this section, we provide a more detailed analysis, presenting a rigorous proof that the guarantees of TRPO are preserved under PR-TRPO and characterize the class of communicators that ensure such stability.

## B.1  GLOBAL SCOPE

Recall, in a metric space of policies $(\Pi, \rho)$, a $\rho$-TRPO algorithm solves the problem (8a) subject to *variational bound*

$$\rho(\pi^k, \pi^{k+1}) \leq \delta. \tag{B.1}$$

Whereas in $\rho$-PR-TRPO each actor's objective (9a) is constrained with *local commuted variational bound*

$$\rho(C_i \pi^k, \pi_i^{k+1}) \leq \varepsilon_i. \tag{B.2}$$

More coarsely, in $\rho$-PR-TRPO, the ensemble of actors is subject to *global commuted variational bound*

$$\rho(C \pi^k, \pi^{k+1}) \leq \varepsilon, \tag{B.3}$$

A natural question arise, may we attain the original bound (B.1) with control of commuted bound (B.2) in order to hold the TRPO trust-region while enabling communication? In this regard, we propose to consider *non-expansive communicators*. Recall

**Definition 0.** An operator $C : \Pi^n \to \Pi^n$ in a metric space $(\Pi, \rho)$[6] is a $\gamma$-*non-expansion* if for all $x, y \in \Pi^n$

$$\rho(Cx, Cy) \leq \gamma \rho(x, y), \tag{B.4}$$

where $\gamma \leq 1$ is the *non-expansiveness coefficient*.

Indeed, in the global scope, given the communication operator $C : \Pi^n \to \Pi^n$ is a $\gamma$-non-expansion, and the global commuted variation (B.3) is controlled at a proper level $\varepsilon$ we may show that

$$\rho(\pi^{k+1}, \pi^k) \leq \rho(\pi^{k+1}, C\pi^k) + \rho(C\pi^k, \pi^k) \leq \delta,$$

---

[6]From now on, we would assume that a metric (norm) on a product space $X^n$ is some product metric (norm) induced by one on $X$.

as since $C$ is a non-expansion,

$$\rho(C\pi^k, \pi^k) \leq \rho(C\pi^k, C\pi^{k-1}) + \rho(C\pi^{k-1}, \pi^k) \leq \gamma\rho(\pi^k, \pi^{k-1}) + \varepsilon, \tag{B.5}$$

it gives

$$\rho(\pi^{k+1}, \pi^k) \leq 2\varepsilon + \gamma\rho(\pi^k, \pi^{k-1}) \leq \delta,$$

as long as we set the *global* commuted variation bound $\varepsilon$ as,

$$\varepsilon \leq \frac{\delta - \gamma\rho(\pi^{k-1}, \pi^k)}{2}.$$

Therefore, *global TRPO variation bound* (B.1) holds for the ensemble of actors and at this point we have proven simple, yet important for heuristic justification of Peer-Referenced methods,

**Observation 0.** *Given a $\gamma$-non-expansion $C : \Pi^n \to \Pi^n$ in a metric space $(\Pi, \rho)$ and a sequence in $\Pi^n$: $\pi^0, \pi^1, \pi^2, \ldots$, the **global variation bound** $\rho(\pi^k, \pi^{k+1}) \leq \delta$ **would hold, given the global commuted variation bound** $\rho(C\pi^k, \pi^{k+1}) \leq \varepsilon$ is held at the level*

$$\varepsilon \leq \frac{\delta - \gamma\rho(\pi^{k-1}, \pi^k)}{2}. \tag{B.6}$$

## B.2 Local scope

However, this tells us nothing about the local behavior of each actor $\pi_i$. Here, a careful choice of an operator family is needed. In time, we would present such a family that, most notably, satisfies a key property of *uniform $\gamma$-non-expansiveness*. Recall

**Definition 1.** *An operator $C : \Sigma^n \to \Sigma^n$ in Banach space $(\Sigma, \|\cdot\|)$ is a uniform $\gamma$-non-expansion $(\gamma \leq 1)$ as long as for $i = 1, \ldots, n$ and for all $x, y \in \Sigma^n$*

$$\|C_i x - C_i y\| \leq \gamma\|x - y\|_\infty, \tag{B.7}$$

*where $\|x - y\|_\infty = \max_i \|x_i - y_i\|$ is a $\ell^\infty$ product norm induced on $\Sigma^n$ by $\|\cdot\|$.[7] Or, in other words, $C$ is $\gamma$-Lipschitz in $(\Sigma^n, \|\cdot\|_\infty)$.*

Now, given the communication operator $C$ is a uniform $\gamma$-non-expansion, the inequality (B.5) may be sharpened for the local scope:

$$\|C_i\pi^k - \pi_i^k\| \leq \|C_i\pi^k - C_i\pi^{k-1}\| + \|C_i\pi^{k-1} - \pi^k\| \leq \gamma\|\pi^k - \pi^{k-1}\|_\infty + \varepsilon.$$

and with reiteration of proof for the global observation 0 we can easily make a local

**Observation 1.** *Given a uniform $\gamma$-non-expansion $C : \Sigma^n \to \Sigma^n$ in a Banach space $(\Sigma, \|\cdot\|)$ and a sequence of vectors in $\Sigma^n$: $\pi^0, \pi^1, \pi^2, \ldots$, the **local variation bound** $\|\pi_i^k - \pi_i^{k+1}\| \leq \delta$ **would hold uniformly, given the local commuted variation bound** $\|C_i\pi^k - \pi_i^{k+1}\| \leq \varepsilon$ **is held uniformly** at the level*

$$\varepsilon \leq \frac{\delta - \gamma\|\pi^{k-1} - \pi^k\|_\infty}{2}. \tag{B.8}$$

Therefore, given the local TRPO variation bound (B.1) is held uniformly at step $k$, i.e. for $i = 1, \ldots, n$

$$\|\pi_i^k - \pi_i^{k+1}\| \leq \delta, \text{ or, equivalently } \|\pi^k - \pi^{k+1}\|_\infty \leq \delta,$$

*the local TRPO variation bound would hold indeed* at the same level $\delta$ for the next step $k + 1$, *given the local PR-TRPO variation bound is controlled uniformly* at the level $\varepsilon$ (12).

The bounds (B.6), (B.8) for the controlled parameter $\varepsilon$ seem feasible in practice, since the right-hand side stays non-zero as long as $C$ is a contraction ($\gamma < 1$) or the desired TRPO variation bound is not reached in norm. However, even if such case takes place we rely upon practical techniques in engineering the exact optimization procedure with use of e.g. scheduling of the parameter $\delta$ or the operator $C$ itself, that, alongside with observations (0) and (1) would provide controllable divergence in TRPO variation. The precise formulation of such designs in theory, accompanied with consequent results we leave out of the scope of this research for the future work as a subject of its own broad field.

---

[7]Although, the discussion may be held in any $\ell^p$ norm induced on $\Sigma^n$, provided $\ell^p$ $\gamma$-non-expansiveness — we would stick with uniform case for clarity.

## B.3 DECOMPOSABLE OPERATORS

Now, as long as we aim to preserve TRPO trust-region, with help of (Observation 1), we need to find the way to construct an appropriate communication operator, that is indeed a uniform non-expansion (Definition 1). In this regard we would consider a class of *decomposable non-expansions*.

**Definition 2.** We would call an operator $C : \Sigma^n \to \Sigma^n$ in Banach space $(\Sigma, \|\cdot\|)$ a *decomposable $\Gamma$-non-expansion*, if for all $x, y \in \Sigma^n$,

$$\|C_i(x) - C_i(y)\| \leq \sum_{j=1}^{n} \gamma_{ij} \|x_j - y_j\|, \tag{B.9}$$

where $\Gamma = (\gamma_{ij}) \in \mathbb{R}^{n \times n}$ is a non-negative non-expansive matrix, i.e. has a spectral radius $\rho(\Gamma) \leq 1$.

*Remark* 1. Given the non-expansive matrix $\Gamma$ of a decomposable "non-expansion" $C$, has the spectral radius $\rho(\Gamma) \leq 1$, there is a product norm induced on the $\Sigma^n$ in which $C$ is indeed a non-expansion.

*Remark* 2. Obviously, a decomposable $\Gamma$-non-expansion is a non-expansion in $\ell^p$ product norm induced on the $\Sigma^n$ *if and only if* $\|\Gamma\|_p \leq 1$, and it follows that $\gamma = \|\Gamma\|_p$ is the non-expansiveness coefficient. We would use these concepts interchangeably from now on.

*Remark* 3. A decomposable $\Gamma$-non-expansion is a uniform $\gamma$-non-expansion if and only if it is a non-expansion in $\|\cdot\|_\infty$ i.e. $\|\Gamma\|_\infty \leq 1$ and $\gamma = \|\Gamma\|_\infty = \max_i \sum_j \gamma_{ij}$.

From a constructive perspective, decomposable non-expansions can be derived using a simple yet sufficiently general class of *decomposable operators*.

**Definition 3.** Given the Banach space $(\Sigma, \|\cdot\|)$ we define a *decomposable operator* $C : \Sigma^n \to \Sigma^n$ component-wise as

$$C_i \pi = \sum_{j=1}^{n} c_{ij}(\pi_j), \tag{B.10}$$

where $c_{ij} : \Sigma \to \Sigma$.

*Remark* 4. A decomposable operator $C : \Sigma^n \to \Sigma^n$ with $\gamma_{ij}$-non-expansive components $c_{ij} : \Sigma \to \Sigma$ s.t. $\rho(\Gamma) \leq 1$, where $\Gamma = (\gamma_{ij})$, is indeed a decomposable $\Gamma$-non-expansion, and thus a non-expansion.

This construction gives us a concrete recipe for building our communication operators: assemble them from non-expansive component maps (Definition 3) whose row-sums remain uniformly bounded. In practice, we often employ the simplest families of such decomposable non-expansions — e.g. matrices $C = \Gamma$ with constant row sums or sparse support — because they're easy to implement and tune.

To make this precise, we leave behind our original metric space $\Pi$ of probability-valued policies and embed into the larger Banach space $\Sigma$ of signed-measure–valued policies, equipped with vector addition and the total-variation norm. There, we require each component map $c_{ij} : \Sigma \to \Sigma$ to be non-expansive, with the sum $C_i(\pi) = \sum_{j=1}^{n} c_{ij}(\pi_j)$ mapping $\Pi^n$ back into $\Pi$. Hence, the overall operator $C : \Pi^n \to \Pi^n$ is a well-defined decomposable non-expansion in $\Sigma^n$, yet its action never leaves the simplex of valid policies.

***Example 1 (Right-stochastic matrices).*** Let $S \in \mathbb{R}^{n \times n}$ be a right-stochastic matrix, i.e., $s_{ij} \geq 0$ and $\sum_{j=1}^{n} s_{ij} = 1$ for all $i$.

It defines an operator $\mathcal{S} : \Sigma^n \to \Sigma^n$ with components

$$\mathcal{S}_i(x) := \sum_{j=1}^{n} s_{ij} x_j,$$

where addition and scalar multiplication are defined in the ambient Banach space $(\Sigma, \|\cdot\|)$.

Since $S$, naturally defines a decomposable $S = \Gamma$-non-expansion with $\|\Gamma\|_\infty^{\text{TV}} = 1$, it is, indeed, a uniform 1-non-expansion in the sense of Definition 1.

It is easy to see that the following classes basically fall into the family of right-stochastic matrices, though we restate them here due to their practical importance.

***Example 2 (Self-preferred mean aggregation).*** To explicitly control the influence each agent assigns to its own learned reference policy versus those of its peers, we define the *self-preferred mean aggregation* operator. Given a self-preference parameter $p \in [0, 1]$, the corresponding right-stochastic matrix $S(p) \in \mathbb{R}^{n \times n}$ is defined component-wise as follows:

$$s_{ij}(p) = \begin{cases} p, & \text{if } i = j, \\ \frac{1-p}{n-1}, & \text{if } i \neq j. \end{cases}$$

This structure enables agents to interpolate continuously between isolated local training and fully peer-focused collaboration:

- For $p = 1.0$, the operator reduces to the identity, representing fully isolated training (e.g., vanilla PPO).

- Intermediate values (e.g., $p = 0.6$) indicate self-focused collaborative training, wherein agents prioritize their own policy but incorporate moderate peer influence.

- Lower values (e.g., $p = 0.2$) yield peer-focused collaborative training, emphasizing peer policies significantly.

We extensively utilize these operators in our experiments on LLM fine-tuning, specifically at levels $p = 1.0$ (isolated training), $p = 0.6$ (balanced collaborative training), and $p = 0.2$ (peer-emphasized collaboration), as in Section 4.2.

Apart from manually selected static weighting, more adaptive communication protocols can be employed — particularly those based on policy similarity or performance signals. These strategies enable agents to preferentially align with peers that are either behaviorally close or empirically successful, as explored in works such as (Wagner et al., 2024). In our experiments, we evaluate two such approaches.

***Example 3 (Dot-product similarity-based communication).*** To encourage communication between behaviorally similar agents, we compute communication weights based on empirical dot products between their action distributions. Specifically, for each agent $i$, the similarity to agent $j$ is defined as

$$s_{ij} \propto \sum_{s \in \mathcal{S}_{\text{ref}}} \langle \pi_i(\cdot \mid s), \pi_j(\cdot \mid s) \rangle,$$

where $\mathcal{S}_{\text{ref}}$ is a fixed reference set of states used for comparison. The inner product $\langle \cdot, \cdot \rangle$ is computed over discrete action distributions, which naturally applies to environments like MiniGrid and language modeling, where the action (or token) space is finite.

The resulting matrix $S = (s_{ij})$ is row-normalized to ensure stochasticity. We use this strategy extensively in MiniGrid environments (Sections 4.1, D.1).

***Example 4 (Reward performance-based communication).*** Agents can assign communication weights based on empirical reward performance. In our Atari experiments (Sections 4.1, D.1), where rewards are non-negative and comparable across agents, we compute weights as:

$$s_{ij} = \frac{r_j}{\sum_{k=1}^{n} r_k},$$

where $r_j$ is the average episodic return of agent $j$ since the last communication round.

This yields a right-stochastic communication matrix that prioritizes alignment with higher-performing peers. While we employ simple linear weighting in our setup, other schemes — such as softmax or rank-based normalization — can be used in environments where reward magnitudes vary more widely or include negative values.

At this point, we have presented a set of simple yet effective communication operators that exemplify the framework. While these constructions are intentionally minimal, our experiments show they can already enhance learning dynamics in practice. More broadly, the space of possible operators is vast — ranging from static heuristics to adaptive protocols based on similarity, reward, inference,

or exploration strategies. This flexibility is where peer-referenced methods are particularly strong: the framework imposes no structural constraints beyond non-expansiveness, ensuring that the TRPO trust region is preserved or degrades controllably. Designing such operators remains an open and promising avenue for both theoretical and applied research.

## C  CONVERGENCE

From the federated optimization perspective, the initial problem (3.1) can be considered as a global objective optimization that aligns with local goals:

$$f^* := \min_{\pi \in \Pi^n} \left\{ f(\pi) := \frac{1}{n} \sum_{i=1}^{n} f_i(\pi_i) = \frac{1}{n} \sum_{i=1}^{n} \mathbb{E}[f_i(\pi_i, \xi_i)] \right\}, \tag{11}$$

where $(\Pi^n, \|\cdot\|)$ is a compact convex subspace of Euclidean space $\Sigma^n$ with a dual space $(\Pi^{n*}, \|\cdot\|_*)$. The classical approach to solving problem (11) is Mirror Descent (MD) (Nemirovsky et al., 1983). It has been investigated for policy optimization in reinforcement learning in many works (Geist et al., 2019; Shani et al., 2019; Neu et al., 2017; Liu et al., 2023). The theory behind MD allows one to create practical and robust RL algorithms; therefore, we propose to analyze the following Algorithm 12:

$$\pi_i^{k+1} = \arg\min_{\pi \in \Pi} \{\gamma \langle \nabla f_i(\pi_i^k, \xi_i^k), \pi \rangle + \mathrm{KL}(\pi, C_i \pi^k)\} \tag{12}$$

The update rule (12) resembles the PR-GRPO objective (5) if we consider the inexact solution of $\arg\min$. The convergence analysis of Algorithm (12) helps to better understand and explain the rationale behind PR-GRPO and PR-PPO(Tomar et al., 2021b). To provide a careful theoretical estimate, we require the following classical assumptions on the target function $f$.

**Assumption 1.** The functions $f_i$ are $L$-smooth, convex and have bounded gradients, i.e., for any $x, y \in \Sigma$ the following inequalities hold: 1. $\|\nabla f_i(x) - \nabla f_i(y)\|_* \leq L\|x - y\|$, 2. $f_i(y) \geq f_i(x) + \langle \nabla f_i(y), y - x \rangle$, 3. $\mathbb{E}\|\nabla f_i(x, \xi)\|_*^2 \leq M^2$.

Now, under these assumptions, we are ready to provide the convergence guarantees for Algorithm 12 in the case of uniform policy averaging through the $C = (c_{ij}) = (1/n)$ communicator.

**Theorem C.1.** *Let Assumption 1 be satisfied. Let problem* (11) *be solved by Algorithm 12. Assume that $\gamma \lesssim \min\left\{ \frac{1}{2L}; \frac{D}{\sqrt{KM}} \right\}$, $D^2 := \max_{x,y \in \Pi} \mathrm{KL}(x, y)$. Then, in order to achieve the $\varepsilon$-approximate solution in terms of $\mathbb{E}[f(\frac{1}{Kn} \sum_{k,i} \pi_i^k) - f(\pi^*)] \leq \varepsilon$ it takes*

$$K = \widetilde{\mathcal{O}}\left( \max\left\{ \frac{LD^2}{\varepsilon} ; \frac{M^2 D^2}{\varepsilon^2} \right\} \right) \text{ iterations of Algorithm (12).}$$

The results of Theorem C.1 align with the convergence rate of classical stochastic Mirror Descent (Nemirovski et al., 2009).

### C.1  CONVERGENCE PROOF

In this section, we provide a convergence proof for Theorem C.1. In the proof, we use the following notation: $\bar{\pi} = C_i \pi^k = \frac{1}{n} \sum_{i=1}^{n} \pi_i^k$. Let also $\omega(x) = -\sum_x p(x) \log p(x)$, and thus $\mathrm{KL}(x, y) = \omega(x) - \omega(y) - \langle \nabla \omega(y), x - y \rangle$.

The following Lemma (we provide the case of KL-divergence) will help us to prove the final result.

**Lemma 1** (Lemma 3 form Juditsky et al. (2011)). *For all $\pi \in \Pi$ we define the prox mapping $P_x(\pi)$ as*

$$P_x(\pi) := \arg\min_{y \in \Pi} \{\mathrm{KL}(y, x) + \langle \pi, y \rangle\}.$$

*For every $x \in \Pi$, the mapping $\pi \mapsto P_x(\pi)$ is Lipschitz continuous, specifically,*

$$\|P_x(\eta) - P_x(\zeta)\| \leq \|\eta - \zeta\|_* \quad \forall \eta, \zeta \in \mathcal{Y}.$$

Now we are ready to provide the proof.

*Proof.* Let us first write the optimality condition for Algorithm 12:

$$\pi_i^{k+1} = \arg\min_{\pi \in \Pi}\{\underbrace{\gamma\langle\nabla f_i(\pi_i^k, \xi_i^k), \pi\rangle + \mathrm{KL}(\pi, \bar{\pi}^k)}_{g(\pi)}\} \Leftrightarrow \forall \pi \in \Pi \hookrightarrow \langle\nabla g(\pi_i^{k+1}), \pi_i^{k+1} - \pi\rangle \le 0.$$

Thus, one can obtain:

$$\langle\gamma\nabla f_i(\pi_i^k, \xi_i^k) + \nabla\omega(\pi_i^{k+1}) - \nabla\omega(\bar{\pi}^k), \pi_i^{k+1} - \pi\rangle \le 0,$$

$$\gamma\langle\nabla f_i(\pi_i^k, \xi_i^k), \pi_i^{k+1} - \pi\rangle \le -\langle\nabla\omega(\bar{\pi}^k) - \nabla\omega(\pi_i^{k+1}), \pi - \pi_i^{k+1}\rangle.$$

Using three point identity, one can get:

$$\gamma\langle\nabla f_i(\pi_i^k, \xi_i^k), \pi_i^{k+1} - \pi^*\rangle \le -(\mathrm{KL}(\pi^*, \pi_i^{k+1}) + \mathrm{KL}(\pi_i^{k+1}, \bar{\pi}^k) - \mathrm{KL}(\pi^*, \bar{\pi}^k)).$$

Now, using straightforward algebra, one can obtain:

$$\gamma\langle\nabla f_i(\pi_i^k, \xi_i^k), \bar{\pi}^k - \pi^*\rangle \le \mathrm{KL}(\pi^*, \bar{\pi}^k) - \mathrm{KL}(\pi^*, \pi_i^{k+1}) - \mathrm{KL}(\pi_i^{k+1}, \bar{\pi}^k)$$
$$+ \gamma\langle\nabla f_i(\pi_i^k, \xi_i^k), \bar{\pi}^k - \pi_i^{k+1}\rangle.$$

$$\gamma\langle\nabla f_i(\pi_i^k), \bar{\pi}^k - \pi^*\rangle \le \mathrm{KL}(\pi^*, \bar{\pi}^k) - \mathrm{KL}(\pi^*, \pi_i^{k+1}) - \mathrm{KL}(\pi_i^{k+1}, \bar{\pi}^k)$$
$$+ \gamma\langle\nabla f_i(\pi_i^k, \xi_i^k), \bar{\pi}^k - \pi_i^{k+1}\rangle + \gamma\langle\nabla f_i(\pi_i^k) - \nabla f_i(\pi_i^k, \xi_i^k), \bar{\pi}^k - \pi^*\rangle.$$

$$\gamma\langle\nabla f_i(\pi_i^k), \pi_i^k - \pi^*\rangle \le \mathrm{KL}(\pi^*, \bar{\pi}^k) - \mathrm{KL}(\pi^*, \pi_i^{k+1}) - \mathrm{KL}(\pi_i^{k+1}, \bar{\pi}^k) + \gamma\langle\nabla f_i(\pi_i^k), \pi_i^k - \bar{\pi}^k\rangle$$
$$+ \gamma\langle\nabla f_i(\pi_i^k, \xi_i^k), \bar{\pi}^k - \pi_i^{k+1}\rangle + \gamma\langle\nabla f_i(\pi_i^k) - \nabla f_i(\pi_i^k, \xi_i^k), \bar{\pi}^k - \pi^*\rangle.$$

Let us define $\delta_i^k = \langle\nabla f_i(\pi_i^k) - \nabla f_i(\pi_i^k, \xi_i^k), \bar{\pi}^k - \pi^*\rangle$. Now, using Cauchy-Schwarz inequality with $\alpha = \gamma^{-1}$:

$$\gamma\langle\nabla f_i(\pi_i^k), \pi_i^k - \pi^*\rangle \le \mathrm{KL}(\pi^*, \bar{\pi}^k) - \mathrm{KL}(\pi^*, \pi_i^{k+1}) - \mathrm{KL}(\pi_i^{k+1}, \bar{\pi}^k) + \gamma\langle\nabla f_i(\pi_i^k), \pi_i^k - \bar{\pi}^k\rangle$$
$$+ \frac{\gamma^2}{2}\|\nabla f_i(\pi_i^k, \xi_i^k)\|_*^2 + \frac{1}{2}\|\bar{\pi}^k - \pi_i^{k+1}\|^2 + \gamma\delta_i^k.$$

Since from 1-strongly convexity of $\omega$ it follows that $-\mathrm{KL}(x, y) \le \frac{1}{2}\|x - y\|^2$, one can get:

$$\gamma\langle\nabla f_i(\pi_i^k), \pi_i^k - \pi^*\rangle \le \mathrm{KL}(\pi^*, \bar{\pi}^k) - \mathrm{KL}(\pi^*, \pi_i^{k+1}) - \frac{1}{2}\|\pi_i^{k+1} - \bar{\pi}^k\|^2 + \gamma\langle\nabla f_i(\pi_i^k), \pi_i^k - \bar{\pi}^k\rangle$$
$$+ \frac{\gamma^2}{2}\|\nabla f_i(\pi_i^k, \xi_i^k)\|_*^2 + \frac{1}{2}\|\bar{\pi}^k - \pi_i^{k+1}\|^2 + \gamma\delta_i^k$$
$$\le \mathrm{KL}(\pi^*, \bar{\pi}^k) - \mathrm{KL}(\pi^*, \pi_i^{k+1}) + \gamma\langle\nabla f_i(\pi_i^k), \pi_i^k - \bar{\pi}^k\rangle$$
$$+ \frac{\gamma^2}{2}\|\nabla f_i(\pi_i^k, \xi_i^k)\|_*^2 + \gamma\delta_i^k.$$

Using Assumption 1, one has:

$$\gamma(f_i(\pi_i^k) - f_i(\pi^*)) \le \mathrm{KL}(\pi^*, \bar{\pi}^k) - \mathrm{KL}(\pi^*, \pi_i^{k+1}) + \gamma(f_i(\pi_i^k) - f_i(\bar{\pi}^k)) + \frac{\gamma L}{2}\|\pi_i^k - \bar{\pi}^k\|^2$$
$$+ \frac{\gamma^2}{2}\|\nabla f_i(\pi_i^k, \xi_i^k)\|_*^2 + \gamma\delta_i^k.$$

$$\gamma(f_i(\bar{\pi}^k) - f_i(\pi^*)) \le \mathrm{KL}(\pi^*, \bar{\pi}^k) - \mathrm{KL}(\pi^*, \pi_i^{k+1}) + \frac{\gamma L}{2}\|\pi_i^k - \bar{\pi}^k\|^2 + \frac{\gamma^2}{2}\|\nabla f_i(\pi_i^k, \xi_i^k)\|_*^2 + \gamma\delta_i^k.$$

Averaging clients from $i = 1$ to $i = n$, one obtains:

$$\gamma(f(\bar{\pi}^k) - f(\pi^*)) \le \mathrm{KL}(\pi^*, \bar{\pi}^k) - \frac{1}{n}\sum_{i=1}^{n}\mathrm{KL}(\pi^*, \pi_i^{k+1}) + \frac{\gamma L}{2}\frac{1}{n}\sum_{i=1}^{n}\|\pi_i^k - \bar{\pi}^k\|^2$$
$$+ \frac{\gamma^2}{2}\frac{1}{n}\sum_{i=1}^{n}\|\nabla f_i(\pi_i^k, \xi_i^k)\|_*^2 + \gamma\frac{1}{n}\sum_{i=1}^{n}\delta_i^k.$$

Knowing that $\mathrm{KL}(\pi, \cdot)$ is convex, one can get using Jensen's inequality:

$$\gamma(f(\bar{\pi}^k) - f(\pi^*)) \leq \mathrm{KL}(\pi^*, \bar{\pi}^k) - \mathrm{KL}(\pi^*, \bar{\pi}^{k+1}) + \frac{\gamma L}{2} \frac{1}{n} \sum_{i=1}^n \|\pi_i^k - \bar{\pi}^k\|^2 \qquad (13)$$

$$+ \frac{\gamma^2}{2} \frac{1}{n} \sum_{i=1}^n \|\nabla f_i(\pi_i^k, \xi_i^k)\|_*^2 + \gamma \frac{1}{n} \sum_{i=1}^n \delta_i^k.$$

Consider the term $\frac{1}{n} \sum_{i=1}^n \|\pi_i^k - \bar{\pi}^k\|^2$. Using Jensen inequality and Lemma 1, one can enroll the sequence:

$$\frac{1}{n} \sum_{i=1}^n \|\pi_i^k - \bar{\pi}^k\|^2 \leq \frac{1}{n^2} \sum_{i=1}^n \sum_{j=1}^n \|\pi_i^k - \pi_j^k\|^2$$

$$= \frac{1}{n^2} \sum_{i=1}^n \sum_{j=1}^n \|P_{\bar{\pi}^k}(\gamma \nabla f_i(\pi_i^{k-1})) - P_{\bar{\pi}^k}(\gamma \nabla f_j(\pi_j^{k-1}))\|^2$$

$$\leq \frac{\gamma^2}{n^2} \sum_{i=1}^n \sum_{j=1}^n \|\nabla f_i(\pi_i^{k-1}) - \nabla f_j(\pi_j^{k-1})\|^2.$$

Now, using Cauchy-Schwarz inequality:

$$\frac{1}{n} \sum_{i=1}^n \|\pi_i^k - \bar{\pi}^k\|^2 \leq \frac{2\gamma^2}{n^2} \sum_{i=1}^n \sum_{j=1}^n \|\nabla f_i(\pi_j^{k-1}) - \nabla f_i(\pi_i^{k-1})\|^2$$

$$+ \frac{2\gamma^2}{n^2} \sum_{i=1}^n \sum_{j=1}^n \|\nabla f_i(\pi_j^{k-1}) - \nabla f_j(\pi_j^{k-1})\|^2$$

$$\leq 2\gamma^2 L^2 \frac{1}{n^2} \sum_{i=1}^n \sum_{j=1}^n \|\pi_i^{k-1} - \pi_j^{k-1}\|^2 + 4\gamma^2 M^2,$$

where in the last inequality we used Assumption 1. Substituting $\gamma \leq \frac{1}{2L}$, we get

$$\frac{1}{n} \sum_{i=1}^n \|\pi_i^k - \bar{\pi}^k\|^2 \leq \frac{1}{2} \frac{1}{n^2} \sum_{i=1}^n \sum_{j=1}^n \|\pi_i^{k-1} - \pi_j^{k-1}\|^2 + 4\gamma^2 M^2,$$

Continuing this process, one might obtain:

$$\frac{1}{n} \sum_{i=1}^n \|\pi_i^k - \bar{\pi}^k\|^2 \leq \left(\frac{1}{2}\right)^k \frac{1}{n^2} \sum_{i=1}^n \sum_{j=1}^n \|\pi_i^0 - \pi_j^0\|^2 + 2\gamma^2 \sigma^2 \sum_{t=0}^{k-1} \left(\frac{1}{2}\right)^t \leq \left(\frac{1}{2}\right)^{k-1} D^2 + 4\gamma^2 M^2,$$

where in the last inequality we used geometric progression. Now, substituting this into the inequality (13), one has:

$$\gamma(f(\bar{\pi}^k) - f(\pi^*)) \leq \mathrm{KL}(\pi^*, \bar{\pi}^k) - \mathrm{KL}(\pi^*, \bar{\pi}^{k+1}) + \frac{\gamma L}{2} \left(\frac{1}{2}\right)^{k-1} D^2 + \gamma^3 M^2 L$$

$$+ \frac{\gamma^2}{2} \frac{1}{n} \sum_{i=1}^n \|\nabla f_i(\pi_i^k, \xi_i^k)\|_*^2 + \gamma \frac{1}{n} \sum_{i=1}^n \delta_i^k.$$

Summing this inequality from $k = 0$ to $K$, one gets:

$$\gamma \left(\frac{1}{K} \sum_{k=0}^K f(\bar{\pi}^k) - f(\pi^*)\right) \leq \frac{1}{K}(\mathrm{KL}(\pi^*, \bar{\pi}^0) - \mathrm{KL}(\pi^*, \bar{\pi}^{K+1})) + \frac{\gamma L D^2}{2K} \sum_{k=0}^K \left(\frac{1}{2}\right)^{k-1} + 2\gamma^3 M^2 L$$

$$+ \frac{\gamma^2}{2} \frac{1}{K} \sum_{k=0}^K \frac{1}{n} \sum_{i=1}^n \|\nabla f_i(\pi_i^k, \xi_i^k)\|_*^2 + \gamma \frac{1}{K} \sum_{k=0}^K \frac{1}{n} \sum_{i=1}^n \delta_i^k$$

$$\leq \frac{D^2}{K} + \frac{2\gamma L D^2}{K} + 2\gamma^3 M^2 L$$

$$+ \frac{\gamma^2}{2} \frac{1}{K} \sum_{k=0}^K \frac{1}{n} \sum_{i=1}^n \|\nabla f_i(\pi_i^k, \xi_i^k)\|_*^2 + \gamma \frac{1}{K} \sum_{k=0}^K \frac{1}{n} \sum_{i=1}^n \delta_i^k$$

Taking the mathematical expectation from both sides, using unbiasedness of $f(\pi, \xi)$ and dividing both sides by $\gamma$, one obtains:

$$\mathbb{E}\left[f\left(\frac{1}{K}\sum_{k=0}^{K}\bar{\pi}^k\right) - f(\pi^*)\right] \leq \frac{2D^2}{\gamma K} + 2\gamma M^2.$$

Now, taking $\gamma$ as $\gamma \lesssim \min\left\{\frac{1}{2L}; \frac{D}{\sqrt{KM}}\right\}$, in order to achieve the $\varepsilon$-approximate solution one should take:

$$K = \widetilde{\mathcal{O}}\left(\max\left\{\frac{LD^2}{\varepsilon} ; \frac{M^2D^2}{\varepsilon^2}\right\}\right)$$

steps of Algorithm 12. This completes the proof. $\qquad\square$

# D    EXPERIMENT DETAILS AND ADDITIONAL RESULTS

## D.1    CLASSICAL RL

In section 4.1 we briefly introduced our extensive experiments in Atari and Minigrid environments to benchmark our PR-PPO algorithm against isolated PPO and federated baseline approaches. Here, we provide experimental setup in more detail.

**Environments.**    To evaluate the performance of our algorithms we selected a diverse set of environments that fall into these two domains:

- **Atari:** We used `AsterixNoFrameskip-v4` and `BeamRiderNoFrameskip-v4`, which represent complex control tasks with high-dimensional pixel observations.
- **Minigrid:** We used `FourRooms-v0`, `DoorKey-6x6-v0`, and `DistShift2-v0`, which represent a variety of navigation and planning challenges.

**Agent Communication.**    In collaborative setups, we considered systems of three agents. The weights in the communication matrix of our collaborative algorithms (both PR-PPO and FedAvg-PPO) are updated during each global communication round proportionally to weighted average of individual agents since the last global communication. We explored three different approaches to communication matrix design:

- Uniform averaging, where each agent contributes equally with $c_{ij} = 1/n$, used in both Atari and Minigrid environments (`PR-PPO-ClassicAvg`), (`FedAvg-PPO-ClassicAvg`).
- Performance-based (4) adaptive averaging, where weights are adjusted based on agent rewards is used specifically in Atari environments (`PR-PPO-WeightedAvg`), (`FedAvg-PPO-WeightedAvg`).
- Similarity-based (3) adaptive averaging, where policy similarity influences the weighting is used specifically in Minigrid environments (`PR-PPO-WeightedAvg`), (`FedAvg-PPO-WeightedAvg`).

**Hyperparameter Search.**    For all algorithms, the configurations were carefully tuned through a grid search over a comprehensive set of hyperparameters (Table 2). Each figure in our results represents the performance of the best configurations, averaged across ten random seeds. It is important to note that during our parameter search, we observed that varying the "Local updates" parameter across different configurations did not result in significant differences in model quality. Therefore, in all subsequent experiments, "Local updates" was fixed at 32 for Minigrid experiments and 16 for Atari experiments. For single-agent setups, we performed grid search over "Total timesteps," "Learning rate," and "KL penalty coeff." In collaborative setups, we focused our hyperparameter search on "Learning rate," "Communication penalty coeff" (for PR-PPO), and "KL penalty coeff" (for FedAvg-PPO). This approach allowed us to isolate the effects of the communication parameters from other aspects of the algorithms.

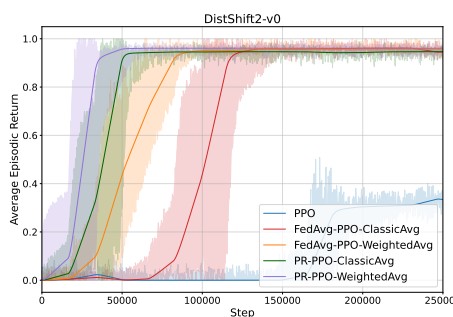 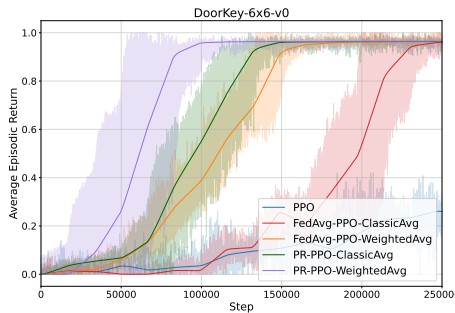

Figure 4: Performance of PR-PPO, isolated PPO and FedAvg-PPO on the `DistShift2-v0`, `DoorKey-6x6-v0` and `AsterixNoFrameskip-v4`. For PR-PPO and FedAvg-PPO the figure shows average performance within the group.

**Computational Resources.** All experiments were conducted on a computing cluster with 15 NVIDIA RTX A4000 GPUs and 152 CPU cores. For Minigrid environments, each experimental run completed within 4-8 hours, while Atari experiments required approximately 24-48 hours per run due to their higher computational demands and longer training durations.

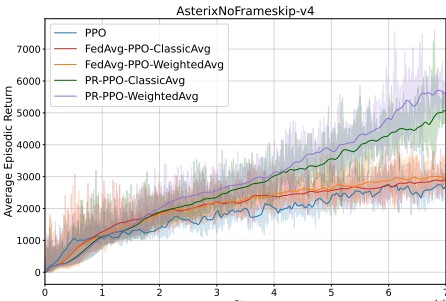

**Additional Results** In addition to experiments presented in the main part (Figure 1), we deliver more results in `DistShift2-v0`, `DoorKey-6x6-v0`, `AsterixNoFrameskip-v4` environments below (Figures 4 and 5) to compare PPO, FedAvg-PPO and our PR-PPO algorithm 4. Within considered domains, our algorithm outperforms both FedAvg-PPO and isolated PPO baselines.

Figure 5: Performance of PR-PPO, isolated PPO and FedAvg-PPO on the `AsterixNoFrameskip-v4`. For PR-PPO and FedAvg-PPO the figure shows average performance within the group.

## D.2 RLHF SUMMARIZATION

**Dataset details.** Following Huang et al. (2024), we initialize the shared reward model from a supervised fine-tuned (SFT) model. To avoid data leakage, we hold out a subset of approximately 16k samples from the full TL;DR dataset for SFT training (referred to as `tldr-sft`) and use the remaining data for PPO training (`tldr-ppo`). A separate preference-labeled dataset, `tldr-preference`, is used to train the reward model.

To construct the heterogeneous TL;DR dataset, we partition the full dataset based on the Reddit subreddit associated with each post, which provides a natural topic label (e.g., *relationships*, *legal advice*, *personal finance*). For experiments involving 7 agents, we assign each agent to the following subreddits:

0: `loseit`

1: `dating_advice`

2: `legaladvice`

3: `offmychest`

4: `personalfinance`

5: `relationship_advice`

6: `tifu`

During PR-PPO and PR-GRPO training, all agents are evaluated on a shared validation split from the `tldr-ppo` dataset. See the size of train and validation subsets in the Table 5.

**Training Parameters.** We provide a detailed overview of the training parameters used in our experimental pipelines. The configuration for supervised fine-tuning (SFT) is shown in Table 3, the reward model training setup in Table 4, and the main training pipeline for PR-PPO and PR-GRPO in Table 5. Following Huang et al. (2024), we initialize the reward model from the SFT checkpoint to improve reward quality. However, models for PR-PPO and PR-GRPO fine-tuning start from the base model checkpoint (without SFT) to isolate the effects of pure RLHF-based alignment introduced by these algorithms.

**Evaluation details.** We compute the winrate by prompting Qwen3-32B Team (2025) with the prompt presented in the Figure 6. This prompt is the analogue of one used in TRL[Judges] library von Werra et al. (2020). We host it on a separate 80GB GPU via vLLM (Kwon et al., 2023) and use the inference parameters presented in Table 6.

**Completion quality overview.** In Figures 7, 8 we present sample model-generated summaries. Across both examples, PR-PPO and PR-GRPO generate summaries that are faithful to the original posts while improving on fluency and completeness compared to their single-agent counterparts. In the breakup scenario, PR-PPO captures key emotional and temporal context ("ex of four years", "received another guitar from friend", "before we stopped talking"), resulting in a more informative and human-like summary. The single-agent PPO output, in contrast, omits relevant relational context and compresses too aggressively, reducing clarity. Similarly, in the financial aid question, PR-GRPO preserves specific numerical detail ("$2,500"), event structure, and the user's uncertainty, whereas the single GRPO summary is more terse and less readable. While the reference summaries provide useful framing, the multi-agent completions often exhibit better structure, nuance, and alignment with the original intent of the posts.

| Category | Parameter | Value |
|---|---|---|
| Environment | Environment type | `atari`, `minigrid` |
| | Gym ID | `AsterixNoFrameskip-v4` `BeamRiderNoFrameskip-v4` `FourRooms-v0` `DistShift2-v0` `DoorKey-6x6-v0` |
| Training | Total timesteps | 5,000,000 (`minigrid`), 10,000,000 (`atari`) |
| | **Learning rate** | $[\mathbf{5 \cdot 10^{-5}, 1 \cdot 10^{-4}, 2.5 \cdot 10^{-4},}$ $\mathbf{5 \cdot 10^{-4}, 1 \cdot 10^{-3}}]$ |
| Algorithms | Value coeff | 0.5 |
| | Entropy coeff | 0.01 |
| | **KL penalty coeff** | **[0.5, 1.0, 2.0, 5.0,** **10.0, 20.0, 50.0]** |
| | Clip coeff | 0.1 |
| | PPO epochs | 4 |
| | Minibatches | 4 |
| Optimization | Anneal learning rate | True |
| | Normalize advantage | True |
| | Max gradient norm | 0.5 |
| GAE | Gamma | 0.99 |
| | Lambda | 0.95 |
| Parallelization Communication | Number of envs | 4 |
| | Number of steps | 512 |
| | **Number of agents** | **[1, 3]** |
| | **Local updates** | **[16, 32, 64, 128]** |
| Agent Communication | **Communication penalty coeff** | **[0.5, 1.0, 2.0,** **5.0, 10.0, 20.0]** |
| | Policy aggregation mode | `PR-PPO-WeightedAvg` `PR-PPO-ClassicAvg` `FedAvg-PPO-WeightedAvg` `FedAvg-PPO-ClassicAvg` |

Table 2: Hyperparameter configuration for classical RL tasks experiments. Parameters in **bold** represent values used in grid search.

| Category | Parameter | Value |
|---|---|---|
| Model | Base model | `Mistral-7B-v0.2` |
| | Model type | `AutoModelForCausalLM` |
| | Torch dtype | `bfloat16` |
| LoRA | Rank / Alpha | 16 / 32 |
| | Dropout | 0.0 |
| | Target layers | `q_proj, k_proj, v_proj, o_proj` |
| | Task type | `CAUSAL_LM` |
| Data | Dataset | `tldr-sft` |
| | Train / Eval size | 16,722 / 1,500 |
| | Max sequence length | 1024 |
| Training | Epochs | 2 |
| | Train / Eval batch size | 4 / 4 |
| | Grad accumulation steps | 4 |
| Optimizer | Optimizer | AdamW |
| | Learning rate | $1 \times 10^{-5}$ |

Table 3: Training configuration for supervised finetuning (SFT) of a Llama 3.2B model using `SFTTrainer` from TRL.

| Category | Parameter | Value |
|---|---|---|
| Model | Base model | `Mistral-7B-v0.2-SFT` |
| | Model type | `AutoModelForSequenceClassification` |
| | Torch dtype | `bfloat16` |
| LoRA | Rank / Alpha | 16 / 32 |
| | Dropout | 0.0 |
| | Target layers | `q_proj, k_proj, v_proj, o_proj` |
| | Task type | `SEQ_CLS` |
| Data | Dataset | `tldr-preference` |
| | Train / Eval size | 92,858 / 2,000 |
| | Max input length | 1024 |
| Training | Epochs | 1 |
| | Train / Eval batch size | 2 / 2 |
| | Grad accumulation steps | 8 |
| | Gradient checkpointing | False |
| Optimizer | Optimizer | AdamW |
| | Learning rate | $1 \times 10^{-5}$ |

Table 4: Training configuration for reward model (RM) on TL;DR using `RewardTrainer` from TRL.

| Category | Parameter | Value |
|---|---|---|
| Model | Model | `Mistral-7B-v0.2` |
| | Number of agents | 7 |
| | Max response length | 58 |
| | Data type | `bf16` |
| | Temperature | 0.7 |
| LoRA (Policy) | LoRA rank / alpha | 64 / 16 |
| | Dropout | 0.0 |
| | Target layers | `q_proj, k_proj, v_proj` |
| | Apply to MLP | True |
| | Apply to Output | False |
| LoRA (Value Model) | LoRA rank / alpha | 16 / 32 |
| | Dropout | 0.0 |
| | Target layers | `q_proj, k_proj, v_proj` |
| | Apply to MLP | True |
| | Apply to Output | True |
| PPO | PPO epochs | 2 |
| | PPO batch size | 32 |
| | Forward batch size | 8 |
| | Grad accumulation steps | 4 |
| | **KL coeff** | **[0.03, 0.1, 0.3]** |
| Data | Dataset | `tldr-ppo` |
| | Train batch / steps | 64 / 26 (1664 samples) |
| | Train epochs | 6 |
| | Eval batch / steps | 8 / 16 (128 samples) |
| GAE (PPO) | Gamma | 1.0 |
| | Lambda | 0.95 |
| | Value coeff | 0.1 |
| | Clip range | 0.2 |
| GRAE (GRPO) | Group size | 8 |
| Reward | Penalize no EOS | True |
| | Penalty value | -3 |
| Reference Model | **Update every N steps** | **[1, 3, 7]** |
| | **Self-preference** | **[0.2, 0.6, 1.0]** |
| Optimizer | Type | AdamW |
| | Learning rate | $1 \times 10^{-4}$ |

Table 5: Training parameters used for PR-PPO and PR-GRPO finetuning of Mistral-7B on the TL;DR task. Parameters marked in **bold** denote values used in grid search for Expected Validation Performance calculation. The main difference between PR-PPO and PR-GRPO is in the advantage estimation method: GAE vs GRAE.

| Parameter | Value |
|---|---|
| `temperature` | 0.6 |
| `top_p` | 0.95 |
| `extra_body` | |
|   `enable_thinking` | True |
|   `top_k` | 20 |
|   `min_p` | 0.0 |

Table 6: Inference parameters for Qwen3 LLM judge.

Prompt template for LLM judge.

```
'''
You are evaluating the performance of different language
models on a summarization task. Each model is given a Reddit
post and tasked with generating a concise and accurate
summary. Your job is to compare the outputs and select the
model that produces the best summary from a human perspective.

## Post

{{
    "post": """\{prompt\}""",
}}

## Model completions

The following are unordered completions with summaries
generated by different models. Each output is associated with
a unique model identifier.

{{
    {{
        "model_identifier": "0",
        "output": """{response0}"""
    }},
    {{
        "model_identifier": "1",
        "output": """{response1}"""
    }}

}}

## Task

Select the model that generated the best summary. Reply with
the identifier of the best model. Our evaluation will only
consider the last character of your answer, so ensure your
response contains only one of the identifiers and nothing
else (no quotation marks, no spaces, no new lines, ...).
'''
```

Figure 6: Prompt template for the LLM judge.

**Reddit post**

SUBREDDIT: r/relationships

TITLE: Do I [21F] have a right to sell something that my ex [21M] of four years gave/sold me upon our breakup?

POST: I had a breakup six months ago with my then-boyfriend of four years. Upon breaking up, we did the old breakup ritual of giving stuff back. He said I could keep a Takamine guitar he had been letting me use, and requested that I only send him a check for $25 for it. I did, I kept the guitar, and that was that.

I thought we parted on good terms, but now we don't talk anymore as he broke off contact with me quite bitterly (he's somewhat emotionally unstable). I do play the guitar on a near daily basis. I don't know how much it's actually worth.

This week, I came home from college, and my best friend just out of the blue *gave* me–for free–a black Ibanez guitar that I absolutely LOVE. More so than the Takamine. For this reason, I'm thinking of selling the Tak–I really don't need two guitars, I could use a little extra money, and I think I have appropriate enough closure on the relationship that I could leave this part of it behind.

My problem is I don't know if it's acceptable to sell something that someone who you no longer have contact with gave you. If it's not acceptable, then I apologize if my question (or I) was rude in any way.

**Reference summary from the dataset**

Ex-boyfriend gave (sold) me a guitar, have recently acquired a better guitar and am thinking of selling the one he gave (sold) to me. What do?

**PR-PPO**

Ex of four years gave me guitar upon breakup, now received another guitar from friend and considering selling first guitar, unsure if it's acceptable to sell something someone gave me before we stopped talking.

**Single PPO**

After breakup ex gave me guitar now im considering selling it, question if its acceptable to sell something given by someone no longer in contact.

Figure 7: Reference summarization sample for PPO.

> **Reddit post**
>
> SUBREDDIT: r/college
>
> TITLE: Are scholarships and grants awarded per semester or year?
>
> POST: I am starting my third semester of college soon and do not quite understand how grants/scholarships work.
>
> My first was in the previous academic school year over the summer and I just completed my second at the start of the current academic year.
>
> From what I've read online it sounds like scholarships and grants are awarded one time a year, but are split up into two parts and refunded to you in those parts at the start of each semester.
>
> So I got $2,500 from just the Pell Grant last semester(which went straight into my bank because scholarships pay for it all), does that mean ill get another check of $2,500 at the start of this upcoming semester or was that first check all i get for the year? The wording on the sites for these grants is a little strange.

> **Reference summary from the dataset**
>
> Are grants split into 2 parts and given to me at separate times throughout the academic year or is all of it given at the start?

> **PR-GRPO**
>
> I got $2,500 refunded to me from grants at the start of last semester, wondering if ill get another check of that amount at the start of upcoming semester or just got all for the year.

> **Single GRPO**
>
> Got grant money last semester, unsure if ill get check w same amount at start of upcoming semester or was that all i get for year from grant.

Figure 8: Reference summarization sample for GRPO.

## E    REPRODUCIBILITY STATEMENT

We have made extensive efforts to ensure the reproducibility of our results. All experiments were conducted using publicly available frameworks and datasets, including Hugging Face TRL library (von Werra et al., 2020), Torchtune (torchtune maintainers and contributors, 2024), and the trl-lib/tldr, trl-lib/tldr-preference datasets. In addition, we release our handcrafted instruments — PrePPO, TunePPO, FedRL, and the anonymous-organization/tldr-thematic dataset — as anonymous supplementary repositories. A detailed description of the learning pipeline and algorithms is provided in the main text, with complete hyperparameter tables (following Dodge et al. (2021)) included in the appendix. Data preprocessing procedures, training configurations, and evaluation protocols are described in detail in the main text and supplementary materials. Together, these resources are intended to facilitate full replication of our results.

## F    LLM USAGE STATEMENT

Large Language Models (LLMs) were used as an assistive tool throughout the preparation of this paper. Specifically, LLMs were employed for text formatting, grammar correction, rephrasing, and improving the clarity and flow of the narrative. They were also used to accelerate literature review by providing preliminary summaries of relevant work, which were subsequently cross-checked against the original sources by the authors. LLMs were *not used* for research ideation, model design, experimental execution, or analysis of results. The authors take full responsibility for the final content of the paper.

