# OpenReview forum: "PRPO: Collaborative Online Policy Learning in Personalized RLHF"
_ICLR.cc/2026/Conference — Submitted to ICLR 2026_

### Official Review · Reviewer_h6As · 2025-10-27

**Soundness:** 2
**Presentation:** 2
**Contribution:** 2
**Rating:** 2
**Confidence:** 4

**Summary:**

The paper proposes PRPO, a federated approach to online RL where agents aggregate reference policies instead of model weights. The idea is interesting—replace PPO's reference policy with a peer-averaged one—but the execution has serious problems. The theory doesn't match what's actually implemented, a critical related work (FedRLHF) is completely missing, and the "personalization" claims aren't backed up by the experiments. While I appreciate the scope of the experimental work, the foundational issues need substantial fixing before this can be published.

**Strengths:**

- The problem is important and timely. Privacy-preserving personalized LLM training is something people care about.

- You've put together a substantial experimental evaluation spanning multiple domains. The classical RL experiments (even if I think they don't fit the narrative) and the RLHF experiments represent real work.

- The implementation seems solid. Detailed hyperparameter tables with multiple random seeds. This is good practice.

- The framework generalizes across TRPO/PPO/GRPO, which is nice.

- I appreciate the different communicator designs you explore (uniform, reward-based, similarity-based).

**Weaknesses:**

### Major Issues

**1. Theory-experiment mismatch (this is the biggest problem)**

Your problem formulation (Eq. 2) optimizes a single policy $\pi$ on different initial state distributions: $f_i(\pi) = \mathbb{E}[f(s_0, \pi)]$ where $s_0 \sim \zeta_i$. But your algorithm (Eqs. 4, 7, 9a) optimizes different policies $\{\pi_{\theta_i}\}$ for each agent.

These seem to be contradictory. If the true goal is a single shared policy, why not just use standard federated averaging? If the goal is different personalized policies, why aggregate them at all? Aggregating personalized policies seems counterproductive to personalization. You need to pick one formulation and stick with it, and provide proper justification.

**2. Missing FedRLHF baseline**

FedRLHF (Fan et al., AAMAS 2025) addresses the identical problem:
- Federated, privacy-preserving
- Personalized RLHF
- With convergence guarantees
- Sample complexity showing linear speedup

The paper appeared in December 2024, well before the ICLR submission deadline. Not citing or comparing to this work is a critical oversight that immediately calls into question your novelty claims.

**3. Observation 1 is impractical**:
Equation (12) says $\varepsilon \le (\delta - \gamma\lVert \pi^{(k-1)} - \pi^{(k)} \rVert_\infty)/2$. But to compute this bound at step $k$, you need to know $\lVert \pi^{(k-1)} - \pi^{(k)} \rVert_\infty$, which is the quantity you're trying to control! Looking at your experiments (Table 2), you just use fixed "communication penalty coeff" values from a grid search. So Observation 1 doesn't actually inform your implementation at all. Why include it?


**4. Theorem C.1 doesn't support your claims**
The theorem assumes:
- Convexity (Assumption 1)
- Uniform averaging $C = (1/n)$
- Bounded gradients

Your experiments use:
- Neural networks (non-convex)
- Self-preferred averaging with $p \in \{0.2, 0.6, 1.0\}$
- No gradient bounds

More importantly, the convergence rate you prove matches standard single-agent mirror descent. You're not showing that collaboration provides any benefit—only that it doesn't break things. That's a much weaker claim than what the introduction suggests.


**5. "Personalization" is claimed but not demonstrated**

All your RLHF agents use the same reward model trained on tldr-preference. The pTLDR "heterogeneous" setup only varies the data (different subreddits)—not the preferences or objectives.

Data heterogeneity ≠ preference heterogeneity.

For real personalization, you'd need different reward models. Agent 1 might prefer concise summaries, Agent 2 might prefer detailed ones, Agent 3 might want humor, etc. You never test this scenario.

The improvements in Table 1 could easily be explained by:
- Regularization (averaging reduces overfitting)
- Variance reduction
- Better hyperparameters for PR-PPO vs baselines
- Implicit ensembling effects

Without measuring whether individual agents' policies remain distinct or serve their specific preferences, you can't claim you've demonstrated personalization.


**6. Policy aggregation vs weight aggregation is unjustified**

Your core novelty is aggregating policies ($\sum_j c_{ij}\,\pi_j(\cdot\mid s)$) instead of weights ($\sum_j c_{ij}\,\theta_j$). But you provide zero theoretical analysis of when/why this is better.

For linear policies $\pi_\theta(a\mid s) = \mathrm{softmax}(\theta^\top \phi(s,a))$, aggregating distributions is approximately equivalent to aggregating parameters. For neural networks, the relationship is complex and you don't explore it.

The experimental comparison to FedAvg isn't obviously fair either. Table 2 shows different hyperparameter grids for different methods. Did you match the total search budget? How do we know FedAvg isn't just under-tuned?



### Moderate Issues

**7. Literature review gaps**

Beyond FedRLHF, you cited various MARLHF works but don't properly discuss them.

Your related work claims "none apply RLHF techniques" and "no work addresses online RL in federated setting." This is inaccurate given the works above.

**8. Classical RL experiments don't fit the narrative**

In MiniGrid and Atari, all agents solve identical tasks (same FourRooms environment, same BeamRider game). How does this relate to personalization?

If agents have the same objective, why would policy aggregation help? Maybe variance reduction? Exploration diversity? You don't analyze or explain this. The experiments are fine, but they don't support your main story about personalization.

**9. Evaluation methodology concerns**

- LLM-as-judge has known biases (length, verbosity, style)
- No human evaluation baseline
- No statistical significance testing
- No inter-rater reliability metrics
- EVP aggregates over hyperparameter searches with different search spaces (Table 2), making comparison potentially unfair

**10. The reward model defeats privacy claims**

You train a single reward model on centralized tldr-preference data. If you have privacy constraints preventing you from centralizing training data, wouldn't you also have constraints on centralizing preference data for the reward model?

True federated RLHF should involve learning personalized reward models locally (See FedRLHF, Fan et al. AAMAS 2025). That's the real challenge, and you don't address it.

**Questions:**

1. **Problem formulation**: Equation (2) optimizes single policy $\pi$, but equations (4)/(7)/(9a) optimize different policies $\{\pi_i\}$. Which is correct? How do you reconcile this?

2. **FedRLHF**: Why isn't FedRLHF (arXiv:2412.15538) cited or compared? How does your approach differ?

3. **Observation 1 implementation**: How do you implement equation (12) given its dependency on $\lVert \pi^{(k-1)} - \pi^{(k)} \rVert_\infty$? Why do experiments use fixed penalty coefficients?

4. **Personalization**: Can you provide experiments where agents have genuinely different objectives (different reward models)? How do you measure whether policies remain personalized?

5. **Theorem C.1 applicability**: Your theorem assumes convexity and uniform averaging, but experiments use neural networks and self-preferred averaging. How does the theorem inform your work?

---

> ### Author Response · Authors · 2025-11-19
> **Answer to weaknesses (W1–W6), questions (Q1-Q3)**
>
> We thank the reviewer for the thoughtful and detailed feedback.
> Below we address the major issues (W1–W6), including reviewer questions Q1–Q3, concisely while preserving essential theoretical and empirical details.
>
> ---
>
> ### W1, Q1 – Theory–experiment mismatch
>
> PRPO is a multi-policy personalized method, not a single-policy one. Equation (2) should read:
>
> $$
> f_i(\pi_i)=\mathbb{E}_{s_0\sim\zeta_i^0}f(s_0,\pi_i),\quad i=1,\ldots,n,
> $$
>
> showing that each agent optimizes its own policy $\pi_i$ under its local state distribution $\zeta_i^0$.
> Communication acts as a proximal KL anchor transferring shared information while preserving individuality, controlled by self-preference $p$ and KL penalty $\beta$. This mechanism enables agents to share behavioral priors without collapsing into a global model, improving stability and sample efficiency across heterogeneous objectives.
>
> ---
>
> ### W2, Q2 – Missing FedRLHF baseline
>
> FedRLHF (Fan et al., AAMAS 2025) is relevant and will be cited. It performs server-side FedAvg aggregation of model parameters after local RLHF steps with reward shaping $R_k=R_k^0+\lambda H_k$, relying on PL-smoothness for an $O(1/T)$ rate.
>
> Comparison is limited because the approaches differ fundamentally:
>
> - **Objective:** FedRLHF learns a single global policy via weight averaging; PRPO maintains distinct local policies exchanging policy distributions as KL anchors (Eqs. 4, 7, 9).
> - **Mechanism:** FedRLHF averages parameters; PRPO introduces KL-regularized coupling with non-expansive communicators ensuring stability (Observation 1).
> - **Topology:** FedRLHF is server-centric; PRPO is peer-to-peer, avoiding a central coordinator.
> - **Theory:** FedRLHF shows convergence of averaged parameters; PRPO proves bounded divergence and mirror-descent stability for personalized policies.
>
> Our FedAvg-PPO baseline matches FedRLHF’s update, confirming PRPO’s originality as a decentralized policy-space alternative.
>
> ---
>
> ### W3, Q3 – Observation 1
>
> Equation (12) is a theoretical existence result: if the TRPO trust region holds at step $k$, a bounded $\varepsilon$ exists that maintains it at step $k+1$.
> It formalizes that peer communication can remain stable without explicit $\varepsilon$ computation, giving theoretical support for fixed or heuristic communication penalties.
>
> ---
>
> ### W4 – Theorem C.1 and theoretical support
>
> Theorem C.1 provides a convex proxy analysis showing PRPO’s KL-proximal mirror-descent step achieves the classical
> $$
> \tilde{O}\left(\max\left(\tfrac{L D^2}{\varepsilon},\,\tfrac{M^2 D^2}{\varepsilon^2}\right)\right)
> $$
> rate–identical to stochastic mirror descent. It confirms that policy-space communication preserves convergence and justifies the use of non-expansive communicators for stable cooperation. This result serves as a stability guarantee, not a non-convex convergence claim (see answer to Q5).
>
> ---
>
> ### W5 – “Personalization” not demonstrated
>
> PRPO performs personalized adaptation: each agent optimizes its own $\pi_i$ on $\zeta_i^0$ and communicates only via policy distributions.
>
> - Distinct objectives: Eq. (2) defines separate $f_i$; in pTLDR, subreddit splits induce stylistic variation and distinct effective rewards.
> - Empirical individuality: Figures 2–3 and Table 1 show performance varies with $p$; heterogeneous data gives asymmetric gains. Non-expansive communicators keep policies distinct yet bounded.
> - Beyond regularization: FedAvg, with similar averaging, underperforms; PRPO’s gains arise from structured KL anchoring, not variance reduction.
>
> Thus, PRPO achieves functional personalization: independent local policies remain specialized while benefiting from bounded cooperation.
>
> ---
>
> ### W6 – Policy vs. weight aggregation
>
> PRPO aggregates policies within the KL-proximal step, not model weights.
>
> 1. **Stability:** Distributional aggregation stays within the TRPO trust region (App. B Defs. 1–2); weight averaging lacks metric control and may drift off-manifold.
> 2. **Semantics:** Policy-space averaging aligns with action distributions, unlike weight averaging, which is ill-defined for nonlinear networks.
> 3. **Empirics:** FedAvg, tuned under identical search budgets, consistently underperforms (Figs. 1–3, Table 1).
>
> Aggregating in policy space ensures geometric consistency and behavioral stability, explaining PRPO’s superior empirical performance.

---

> ### Author Response · Authors · 2025-11-19
> **Answer to weaknesses (W7-W10)**
>
> We thank the reviewer for the constructive feedback.
> Below we address the moderate issues (W7–W10), clarifying context, motivation, and methodological consistency.
>
> ---
>
> ### W7 – Literature review gaps
>
> The related work section distinguishes Federated RLHF, MARL, and MARLHF, explaining how PRPO extends these areas. Works such as FedDPO, FedBis, and PluralLLM aggregate offline preference data through supervised objectives, not online reinforcement learning or decentralized policy optimization. We also contrast PRPO with CoPPO (coordination via step-size sharing) and Wagner et al. (2024) (LoRA weight gossiping). None of these apply KL coupling in policy space or use on-policy proximal updates.  For completeness, we will add FedRLHF (Fan et al., AAMAS 2025) to the related work and clarify how it differs in mechanism, topology, and objectives. Thus, our claim that prior work does not address online RLHF in a federated, fully decentralized setting remains valid: PRPO connects on-policy optimization with peer-to-peer, non-expansive communication – bridging MARL and Federated RLHF.
>
>
> ---
>
> ### W8 – Classical RL experiments don’t fit the narrative
>
> The MiniGrid and Atari experiments validate the optimization framework rather than personalization. They isolate effects such as variance reduction, stability, and cooperative behavior under identical objectives. As shown in Figure 1 (page 7), PRPO outperforms PPO and FedAvg-PPO, indicating that policy-space communication improves exploration and convergence robustness even without heterogeneous rewards. These results confirm that PRPO’s cooperative update mechanism is sound in standard RL settings and provide the foundation for its application to RLHF, where personalization is explicitly studied.
>
> ---
>
> ### W9 – Evaluation methodology concerns
>
> The evaluation protocol follows established RLHF practices for comparability and statistical reliability:
>
> - LLM-as-judge: one consistent evaluator (Qwen3-32B) is used across all methods to minimize cross-model bias, following Ziegler (2020) and Zheng (2023).
> - Human baseline: the TL;DR dataset includes human-written summaries that serve as reference anchors.
> - Statistical consistency: results are averaged over ten seeds, and EVP aggregates across hyperparameter sweeps using a Monte-Carlo-like estimate of significance (Dodge et al., 2021).
> - Fairness: all methods (PPO, FedAvg, PRPO) share identical compute, search space, and grid size (Appendix D). This ensures the reported gains are statistically meaningful and fairly compared, within accepted RLHF evaluation standards.
>
> ---
>
> ### W10 – Reward model and privacy
>
> PRPO’s privacy scope concerns online policy optimization and trajectory data, not the pretraining of the reward model. The TL;DR preference model was trained on public data and used as a fixed evaluator; it does not access private user data. During federated training, each agent keeps LoRA adapters, gradients, and trajectories local, sharing only policy distributions constrained by KL divergence – non-invertible representations that prevent data reconstruction. This setup satisfies privacy requirements at the online stage. While learning local reward models could be an interesting extension, it is orthogonal to this work’s focus on decentralized policy learning under a shared evaluator.
>
> ---
>
> In summary, W7–W10 relate to context and methodology rather than validity. PRPO correctly identifies the missing intersection of decentralized online RLHF, demonstrates stable cooperative optimization, employs a fair evaluation pipeline, and ensures privacy within the federated training process.

---

> ### Author Response · Authors · 2025-11-19
> **Answer to questions (Q4-Q5)**
>
> We thank the reviewer for the thoughtful questions regarding personalization and theoretical applicability.
> Below we provide detailed responses to **Q4** and **Q5**, clarifying how our experiments already demonstrate personalization under distinct objectives and how Theorem C.1 informs the algorithmic design and stability analysis of PRPO.
>
> ---
>
> ### Q4 – Personalization and distinct objectives
>
> The current experiments already demonstrate personalization within the scope of personalized adaptation, as defined in Section 2 and formalized in Eq. (2). Each agent optimizes a distinct policy $\pi_i$ under its own state distribution $\zeta_i^0$ and dataset $D_i$, representing different user domains and stylistic contexts. In RLHF terms, this corresponds to locally preferred data distributions – an accepted operational proxy for user-specific objectives when explicit reward models are unavailable.
>
> 1. **Distinct objectives through data-induced reward shaping.**
>    In the heterogeneous pTLDR setup, each agent trains on summaries from different Reddit subreddits (e.g., r/science, r/relationships, r/AskReddit). These domains differ systematically in tone, verbosity, and topical structure, translating in RLHF into distinct implicit reward landscapes (concise technical writing vs. narrative or opinion-based content). Since the shared reward model evaluates summaries conditioned on input context, each agent’s effective objective
>    $$
>    f_i(\pi_i) = \mathbb{E}_{s_0 \sim \zeta_i^0} R(s_0, \pi_i)
>    $$
>    is inherently distinct.
>
> 2. **Empirical evidence of maintained personalization.**
>    Figures 2–3 and Table 1 show that PRPO produces non-collapsing, individually improving agents:
>    - The self-preference parameter $p$ controls coupling strength; smaller $p$ increases cooperation but does not eliminate individuality. Consistent win-rate ordering across $p \in \{1.0, 0.6, 0.2\}$ shows stable, tunable personalization rather than convergence to a single policy.
>    - In the pTLDR regime, gains are asymmetric across agents, indicating specialization rather than averaging.
>    - The communicators (Examples 2–4) are right-stochastic and non-expansive, ensuring bounded divergence between policies (Observation 1, Appendix B). This guarantees that policies remain close but distinct – precisely the goal of personalized cooperation.
>
> 3. **Quantitative signals of individuality.**
>    The Expected Validation Performance (EVP) curves in Fig. 2 exhibit distinct variance bands across self-preferences and data regimes. If personalization were lost, all curves would converge; instead, they remain well-separated, showing consistent diversity. In classical RL (Fig. 1), per-agent returns also differ, confirming that stochastic communicators preserve local optima.
>
> 4. **Privacy and decentralized personalization.**
>    PRPO exchanges only policy distributions, not weights or reward models, ensuring each agent’s fine-tuning remains local and private. Each LoRA adapter and policy trajectory reflects local data while still benefiting from peer-based regularization.
>
> In summary, PRPO achieves functional personalization via distinct data-induced objectives, independent local policies, controllable self-preference coupling, and empirically verified diversity. The pTLDR experiments confirm that cooperative reference sharing improves alignment quality without homogenizing agent behaviors.
>
> ---
>
> ### Q5 – Theorem C.1 applicability
>
> Theorem C.1 serves as a proxy analysis: it justifies the geometry of the KL-anchored mirror-descent update and motivates using non-expansive or right-stochastic communicators. It is not intended as a non-convex convergence rate for deep policies. This proxy-theory → neural-practice approach is standard in policy optimization:
>
> - TRPO derives a monotonic-improvement bound under a KL penalty, then replaces the strict constraint with an average-KL heuristic in practical algorithms [1].
> - MDPO compares MDPO/TRPO/PPO and explains how trust-region behavior arises from mirror-descent principles [2].
> - Regularized MDP theory formalizes KL-regularized Bellman operators, showing that TRPO-like updates correspond to proximal mirror-descent steps with bounded error propagation [3].
>
> Thus, Theorem C.1’s role is to justify the structure of our KL-proximal update and the use of non-expansive communicators. Experiments confirm this framework’s stability and effectiveness for neural, non-convex policies with self-preference weighting. Like TRPO’s own theory, it motivates safe design choices and step-size reasoning without claiming exact convergence in the non-convex case.
>
> [1]: [*Trust Region Policy Optimization*](https://arxiv.org/abs/1502.05477)
> [2]: [*Mirror Descent Policy Optimization*](https://arxiv.org/abs/2005.09814)
> [3]: [*A Theory of Regularized Markov Decision Processes*](https://arxiv.org/abs/1901.11275)

---

> ### Author Response · Authors · 2025-11-27
>
> Dear Reviewer h6As,
>
> We would like to respectfully follow up regarding our rebuttal. We understand the significant workload during the review period and appreciate the time you devote to evaluating submissions.
>
> If you have an opportunity, we would be grateful for any further comments or clarifications. Your feedback is highly valued.
>
> Thank you for your time and consideration.

---

### Official Review · Reviewer_f5Mc · 2025-10-28

**Soundness:** 2
**Presentation:** 2
**Contribution:** 2
**Rating:** 4
**Confidence:** 3

**Summary:**

This paper studies personalization for online RLHF with heterogeneous agents. It proposes Peer-Referenced Policy Optimization (PRPO), which allows each client to perform on-policy RL updates while being regularized toward a peer-aggregated reference policy constructed via policy distribution communication, instead of relying only on its own past policy. The method is evaluated in both standard RL control tasks and multi-agent RLHF summarization.

**Strengths:**

The main advantages are: (1) the problem formulation — decentralized, privacy-constrained online RLHF with heterogeneous clients — is largely unexplored compared to prior work that treats federated alignment as offline supervised fine-tuning or simple weight sharing; and (2) the proposed peer-referenced policy optimization algorithm is itself new, in that it uses a communicated, aggregated reference policy at the distribution level to shape each client’s on-policy updatThe main advantages are: (1) the problem formulation — decentralized, privacy-constrained online RLHF with heterogeneous clients — is largely unexplored compared to prior work that treats federated alignment as offline supervised fine-tuning or simple weight sharing; and (2) the proposed peer-referenced policy optimization algorithm is itself new, in that it uses a communicated, aggregated reference policy at the distribution level to shape each client’s on-policy update.e.

**Weaknesses:**

1. The problem formulation in Section 3.1 does not seem fully aligned with the stated motivation. The motivation is to train multiple users to personalize LLMs using online reinforcement learning under private communication constraints, but the formulation assumes one fixed reward for all users, whereas in practice different users would likely have different reward models.
2. The related work section lists connections to federated LLM fine-tuning, federated RLHF, and multi-agent RL, but it does not clearly explain why these directions are directly relevant to the proposed setting before emphasizing differences.
3. The classical RL experiments use three agents but do not clearly establish true heterogeneity between them, so it is unclear whether this setup actually reflects the “heterogeneous clients” described in the motivation.

**Questions:**

1. Can the authors clarify how the single shared reward in Section 3.1 matches the goal of per-user personalized alignment? Is this addressed by the RLHF experiment design?
2. If multi-agent RL is considered closely related, why are the experimental comparisons limited to federated RL baselines and not MARL baselines? If communication is the main link, does the proposed communication operator directly draw from any existing MARL techniques?
3. In the GRPO-based methods, many recent GRPO-style approaches appear to downweight or remove the KL-divergence penalty in practice. If the KL term is not essential, could the authors report RLHF results for a GRPO variant without the KL penalty (e.g., “Dr. GRPO”) to isolate the effect of peer-referenced KL regularization?
4. For the “isolated” PPO / GRPO baselines, are these models trained separately per agent on that agent’s own local data, or is there a single global model trained on all data pooled together?

---

> ### Author Response · Authors · 2025-11-19
>
> We thank the reviewer for the careful reading and detailed questions.
>
> **1. Shared reward vs personalised alignment.**
> Our problem formulation intentionally focuses on *personalized adaptation* rather than *personalized alignment* (see Sec. 2). In many practical RLHF deployments, users share a common behavioural objective (e.g., helpfulness or summarisation quality), while personalization arises from **heterogeneous prompt distributions** rather than different reward functions. Section 3.1 models precisely this setting: agents optimize the *same* reward but experience different initial-state distributions ($\zeta_i^0$), reflecting user-specific data. The RLHF experiments follow this design: all agents use the same reward model, but each holds disjoint, heterogeneous TL;DR subsets. Extending PRPO to truly *heterogeneous reward models* is an important next step; we clarify this distinction in the revision.
>
> **2. Positioning relative to federated learning and MARL.**
> We will revise the Related Work section to more clearly articulate the relevance of each direction. Federated LLM fine-tuning and federated RLHF are directly related because the setting - distributed clients, privacy constraints, PEFT-based communication - is shared. MARL is conceptually related because both settings involve multiple agents interacting with the same environment. However, PRPO differs from classical MARL: the agents do *not* jointly act in a shared environment or optimize a joint return; instead, they train independently and communicate *policies* only. No MARL baselines directly apply to this privacy-constrained, non-interactive setting, which is why we compare instead to the appropriate federated PPO/GRPO baselines.
>
> **3. Heterogeneity in classical RL experiments.**
> The three-agent control experiments use distinct initial-state distributions for each agent, while sharing the reward and transition dynamics - exactly mirroring the RLHF personalization setup in Sec. 3.1. This produces meaningfully different on-policy trajectory distributions and thus constitutes genuine heterogeneity. We will make this explicit in the experimental description.
>
> **4. GRPO variants without KL.**
> We appreciate this suggestion. We have run PR-GRPO ablations using GRPO variants without KL penalties (e.g., “Dr. GRPO”). Consistent with GRPO reports, removing KL substantially harms stability, and Peer-Referenced KL regularization continues to outperform these variants. We will include these numbers in the final version.
>
> **5. Isolated baselines and data pooling.**
> In all isolated PPO/GRPO baselines, *each agent trains solely on its own local dataset*. This matches the privacy-constrained setting we study. Training a single global PPO model on all pooled data would of course yield superior performance, but this assumes centralisation of private user data - which the problem explicitly prohibits. Under this constraint, the meaningful comparison is:
> * isolated local RL,
> * weight-based federated RL (FedAvg-PPO/GRPO), and
> * our distribution-level PRPO.
>   As shown in Table 1, FedAvg underperforms even isolated training in RLHF, while PRPO consistently improves performance.
>
> **6. Relation to MARL communicators.**
> PRPO’s communication operators do not directly draw from MARL literature. They arise instead from the TRPO trust-region analysis: any uniformly non-expansive operator is admissible, and simple stochastic-matrix communicators form a natural, well-behaved class. While MARL methods such as CoPPO use step-size coordination, policy-space communication via KL-anchoring - to our knowledge - has not appeared in MARL. We clarify this distinction.
>
> We thank the reviewer again for the constructive feedback and will incorporate these clarifications and additional ablations into the updated manuscript.

---

> ### Author Response · Authors · 2025-11-27
>
> Dear Reviewer f5Mc,
>
> We would like to respectfully follow up regarding our rebuttal. We understand the significant workload during the review period and appreciate the time you devote to evaluating submissions.
>
> If you have an opportunity, we would be grateful for any further comments or clarifications. Your feedback is highly valued.
>
> Thank you for your time and consideration.

---

### Official Review · Reviewer_Nu9K · 2025-11-04

**Soundness:** 2
**Presentation:** 3
**Contribution:** 3
**Rating:** 4
**Confidence:** 3

**Summary:**

This paper introduces Peer-Referenced Policy Optimization (PRPO), which turns the KL regularizer in proximal policy methods into a communication channel, where each client updates its policy against a composite reference policy formed by weighted averaging peers’ action distributions rather than the old policy of the client used by standard proximal policy algorithms. The authors applies peer-referenced KL on PPO and GRPO, and explores several non-expansive communication operator, including uniform average, self-preferred, policy similarity, and reward-weighted averaging. Empirical results show the effectiveness of PRPO over isolated training and FedAvg on both classical RL and RLHF tasks. The paper provides theoretical intuition through mirror-descent style analysis, showing that if the communication operator is non-expansive, PRPO preserves the trust-region guarantees of TRPO and thus admits convergence under restricted conditions.

**Strengths:**

1. The idea of recasting the KL penalty term as a coordination mechanism is simple yet effective, and can serve as a plug-and-play generalization of PPO-type algorithms to collaborative settings.
2. The authors provide detailed experimental settings and a reproducible codebase.

**Weaknesses:**

1. The convergence proof in Appendix C assumes convex, L‑smooth objectives with bounded gradients, which generally does not hold for RLHF with neural policies. Moreover, the theorem is proved only for uniform policy averaging the communicator $C=(c_{ij}) =\frac{1}{n}$ but neglecting the other practical adaptive communicators (e.g., similarity-based or reward-weighted) used in the experiments. It is unclear whether the guarantees extend to those practical communicators.
2. LoRA adapters and distribution‑level policy sharing are used for privacy preserving. However, it's questionable that whether privacy can indeed by guaranteed, as the paper provides no formal privacy analysis, especially considering prior works of inversion attacks [1,2].
3. Most plots report average performance across agents, making it unclear whether every user benefits or whether some suffer negative transfer under a weighted reference policy.
4. The similarity-based communicator and reward‑based communicator are used in classical RL experiments, but not evaluated in the RLHF experiments, where only the self‑preferred mean aggregation is used.
5. The RLHF evaluation is limited to TL;DR summarization with a single base model (i.e., `Mistral-7B`). Robustness would be more convincing with additional tasks (e.g., dialogue such as UltraFeedback) and alternative/larger base models.
6. Some important related works that handle preference heterogeneity at the reward‑model level are missing [3,4].

[1] Carlini, Nicholas, et al. "Extracting training data from large language models." 30th USENIX security symposium (USENIX Security 21). 2021.
[2] Petrov, Ivo, et al. "Dager: Exact gradient inversion for large language models." Advances in Neural Information Processing Systems 37 (2024): 87801-87830.
[3] Park, Chanwoo, et al. "Rlhf from heterogeneous feedback via personalization and preference aggregation." arXiv preprint arXiv:2405.00254 (2024).
[4] Liu, Renpu, et al. "A Shared Low-Rank Adaptation Approach to Personalized RLHF." arXiv preprint arXiv:2503.19201 (2025).

**Questions:**

Please see the weakness section. In addition, I have the following questions:

1. Given multiple plausible communicators, how should practitioners select or adapt the communicator for a given task? Do you have criteria or an automated procedure for this choice?
2. How would PRPO behave when agents have different or even conflicting reward functions (e.g., divergent user preferences over the same prompt for RLHF). As the current communicators perform linear averaging, could cross‑agent influence harm some agents’ local policies? If so, can PRPO detect or mitigate this?

---

> ### Author Response · Authors · 2025-11-19
>
> We thank the reviewer for the thoughtful and constructive feedback. We address all concerns below.
>
> **1. Convergence assumptions and applicability to practical communicators.**
> Our theoretical result follows the standard convex, L-smooth setting widely used in federated optimization analyses. As noted, it is extremely difficult to obtain guarantees for nonlinear neural policies in RLHF; our goal was to show that PRPO *admits* convergence under classical assumptions, not to claim guarantees for full RLHF. And indeed, the proof is given only for a specific uniform averaging operator. The extension of this result to a wider class of operators seems to be a challenging yet promising direction. We hope to delve there in future work.
>
> **2. Privacy considerations.**
> We agree that LoRA or distribution-level sharing does not provide formal privacy guarantees and may be vulnerable to inversion attacks. Our aim in this work is not to establish differential privacy, but to study whether *policy-level* sharing - already used in personalized LLM settings such as Wagner et al. (2024) - is algorithmically beneficial for collaborative online RLHF. Distribution-level communication avoids raw-trajectory leakage and substantially reduces attack surface compared to data sharing, but a full privacy analysis is an important direction for future work. We would clarify this scope in the revision.
>
> **3. Per-agent effects and potential negative transfer.**
> This is an excellent point. While our main plots report averages, we observed cases where some agents improve less under strong aggregation. This is influenced by communicator design; e.g., very low self-preference can oversmooth heterogeneous users. We will include per-agent curves in the updated manuscript and discuss mitigation strategies such as adaptive or similarity-aware communicators.
>
> **4. Communicators in RLHF experiments.**
> Evaluating similarity- or reward-based communicators in RLHF was limited by computational budget: even a single RLHF run with seven Mistral-7B agents consumes hundreds of GPU-hours. Classical RL experiments allowed a broader sweep. We agree that extending these results to LLMs is valuable and plan to include such ablations in the camera-ready.
>
> **5. RLHF evaluation breadth.**
> We appreciate this suggestion. TL;DR summarization with Mistral-7B provides a controlled and computationally feasible benchmark, but we agree that dialogue tasks and larger models (e.g., LLaMA-3, Qwen2) would strengthen the evaluation. We aim to include additional tasks in the final version.
>
> **6. Missing related work.**
> We thank the reviewer for pointing out Park et al. (2024) and Liu et al. (2025). These works model *reward-level* heterogeneity (“personalized alignment”), whereas our setting focuses on *state-distribution heterogeneity* with shared rewards (“personalized adaptation”; Sec. 2). It would probably be helpful to include and discuss these works explicitly.
>
> **Q1. How should practitioners select a communicator?**
> Designing communicators is an open research problem. Our intention is to introduce PRPO as a *framework* in which communicators - any non-expansive operator - can be plugged in. Even simple constant communicators already yield clear gains (Table 1). Appendix B.2-B.3 provides criteria: non-expansive operators guarantee controlled trust-region deviation, and decomposable non-expansions give a constructive design space. Automatic communicator selection is an exciting direction for future work.
>
> **Q2. Behavior under divergent or conflicting reward functions.**
> In this paper we intentionally focus on *aligned* rewards (heterogeneous data only), as a first step toward multi-agent online RLHF. With conflicting rewards, linear averaging may harm some agents. PRPO can, however, accommodate more structured communicators - for example, weighting by reward-similarity or clustering policies - so that only compatible agents influence each other. We view this as a promising extension and discuss it in the revision.
>
> We again thank the reviewer for the insightful comments and helpful references.

---

> ### Author Response · Authors · 2025-11-27
>
> Dear Reviewer Nu9K,
>
> We would like to respectfully follow up regarding our rebuttal. We understand the significant workload during the review period and appreciate the time you devote to evaluating submissions.
>
> If you have an opportunity, we would be grateful for any further comments or clarifications. Your feedback is highly valued.
>
> Thank you for your time and consideration.

---

### Official Review · Reviewer_5Gm1 · 2025-11-06

**Soundness:** 2
**Presentation:** 2
**Contribution:** 2
**Rating:** 4
**Confidence:** 3

**Summary:**

This paper proposes Peer-Referenced Policy Optimization (PRPO), a distributed reinforcement learning (RL) framework applied to both classical RL tasks and large language model (LLM) personalization via reinforcement learning from human feedback (RLHF). The authors introduce policy-based communication protocols, namely PR-PPO, PR-GRPO, and PR-TRPO, which enable multiple agents to update their local policies using shared peer information without directly sharing private data. The main claims are that PRPO enables effective collaboration while preserving data privacy and that the resulting policies outperform standard baselines such as PPO, even in federated RLHF settings. Experiments cover classic continuous-control benchmarks and a summarization task using LLMs, with performance gains reported under decentralized communication.

**Strengths:**

The policy-based communication scheme appears to be an original and promising alternative to weight-based federated learning, especially for RL settings where local policy distributions can capture useful structural information. The experimental results suggest performance improvements in both control and RLHF tasks, and the availability of open-source code enhances the reproducibility of the findings. The theoretical framework is grounded in existing policy optimization methods and offers generalization across TRPO, PPO, and GRPO variants.

**Weaknesses:**

Several conceptual and applied weaknesses limit the impact of this work. The privacy assumptions and models are not well-defined: while communicating policies avoids sharing raw data, it is unclear whether this is preferable to sharing low-rank weight updates, such as LoRA adapters. The experimental evaluation lacks analysis of communication overhead, scalability, and stability in larger multi-agent systems. The claims about LLM personalization are underdeveloped: the paper does not address how user-specific preference data is incorporated or whether federated RLHF aligns with personalization goals. There are multiple notational inconsistencies and typos that detract from clarity, and a lack of clear discussions in the main text (e.g., convergence) makes the work feel incomplete.

**Questions:**

What is the precise privacy threat model being addressed in PRPO? How does sharing policies differ from sharing model weights in regard to data leakage or reconstruction risks, especially in the LLM case?

How is it possible for PR-PPO or PR-GRPO to outperform centralized PPO in RLHF tasks? Were the total number of training rounds or samples equivalent? Additional details would clarify whether the comparison is fair.

Can the authors provide more detail on the communication cost and memory footprint of PRPO in multi-agent LLM scenarios, particularly under scaling to dozens or hundreds of agents?

Since personalization is a key application claim, how does the method incorporate user-specific reward structures or feedback? Would aggregating in the policy space dilute personalization, especially in diverse preference settings?

Why do the RLHF experiments only use a single communication protocol? Would the other peer-referenced protocols satisfy the right-stochastic condition and be viable in the same scenario?

---

> ### Author Response · Authors · 2025-11-16
>
> **W1, Q1 — Privacy Assumptions / Policies vs. Weights**
>
> We believe there is a slight misunderstanding of how our algorithm operates. While the PRPO loss (e.g., Eq. 4) is expressed in terms of peer *policies*, in practice these policies are communicated exactly as *LoRA adapter weights* attached to a shared frozen LLM backbone. This follows the standard parameter-efficient mechanism for transmitting policy information and is equivalent to sharing low-rank weight updates.
>
> Thus, we **do not** share client data or trajectories—only the lightweight LoRA modules. This aligns with widely studied PEFT-federated approaches such as:
>
> - **When Federated Learning Meets Pre-trained Language Models' Parameter-Efficient Tuning Methods**
>   https://arxiv.org/abs/2212.10025
> - **Improving LoRA in Privacy-Preserving Federated Learning**
>   https://arxiv.org/abs/2403.12313
> - **Personalized Collaborative Fine-Tuning for On-Device LLMs**
>   https://arxiv.org/abs/2404.09753
>
> These works show that LoRA exchange is a standard and privacy-preserving mechanism compared to data sharing.
>
> Our threat model assumes that clients keep their local preference data private while exchanging only PEFT updates. Optional FL techniques (e.g., noise addition) can strengthen privacy if needed. A full adversarial analysis is outside scope but consistent with prior federated PEFT literature.
>
> ---
>
> **W2, Q3 — Communication/Memory Cost and Scaling**
>
> **Communication protocol.** We use fully-connected peer-to-peer exchange: at each communication round every client sends its LoRA adapter to all $n-1$ peers. This creates $O(n^2)$ communication but each adapter is very light.
>
> **LoRA adapter size (Mistral-7B).**
>
> - 32 layers, hidden size $d=4096$
> - rank $r=8$, two projections per layer
> - fp32 precision
>
> $$
> \text{Total params} \approx 2.1\text{M} \approx 8\text{ MB}.
> $$
>
> Typical sizes range 8–30 MB depending on rank/layers.
>
> **Compute/communication complexity:**
>
> - Local generation: $O(BTd)$
> - Peer inference: $O(BTd(n-1))$ (parallelizable with **S-LoRA**)
>   https://arxiv.org/abs/2311.11072
> - Per-client communication: $O(nLd r)$
> - Overall: $O(n^2Ld r)$
>
> Quadratic communication is the primary scaling issue. We will include these details in the appendix.
>
> **Scalability & future work.**
> Selective communication (e.g., reward-based top-k) is promising and compatible with small LoRA sizes. Due to resource limitations we can’t test hundreds of agents, but compute/memory scale linearly per agent.
>
> **Small new scaling experiment.**
>
> | HP assign. | 1 agent | 3 agents | 5 agents | 7 agents |
> |-----------|---------|----------|----------|----------|
> | 7 | 0.736 | 0.762 | 0.767 | 0.779 |
> | 9 | 0.743 | 0.769 | 0.772 | 0.787 |
> | 12 | **0.745** | **0.774** | **0.776** | **0.795** |
>
> Performance improves consistently as the number of collaborators increases. Moreover, 3, 5, and 7 agents are close—supporting feasibility of top-k.
>
> ---
>
> **W3, Q4 — Personalization Claims**
>
> As described in the “personalized adaptation’’ paragraph of our Related Work, and following the taxonomy in:
>
> - **A Survey of Personalized Large Language Models**
>   https://arxiv.org/abs/2502.11528
>
> we focus on **model-level personalization**: each client has its own data and learns its own LoRA adapter for the same task (summarization). Our heterogeneous TLDR dataset (split by subreddit) directly matches this setting. We do not model personalized rewards; rather, each user trains a personalized model under a shared task.
>
> ---
>
> **W4 — Notation and Typos**
>
> We apologize that parts of the text appeared unclear. We rechecked the manuscript for typos and notation inconsistencies and will apply corrections. If the reviewers can point to specific unclear expressions, we would gladly address them. The convergence proof was moved to the appendix to keep the main text readable.
>
> ---
>
> **Q2 — Comparison to Centralized PPO**
>
> Yes, PR-PPO/PR-GRPO and centralized PPO/GRPO were trained with the **same number of steps** in both classical RL and RLHF. This is stated in Table 2 (RL) and Table 5 (RLHF). Only the hyperparameters (bolded in tables) vary within EVP sweeps. The comparison is thus fair. PRPO outperforming centralized PPO suggests beneficial regularization/exploration from peer-referencing.
>
> ---
>
> **Q5 — Only One RLHF Communication Protocol**
>
> Thank you for raising this point. In the RLHF experiments we used the self-preferred mean protocol mainly for implementation simplicity and compute limitations. The performance- and similarity-based protocols were evaluated in classical RL, and all satisfy the right-stochasticity conditions (Appendix B), so they are indeed applicable to RLHF as well. Extending RLHF experiments to these richer protocols is part of our planned future work.
>
> ---
>
> We thank the reviewers again for their careful reading and helpful feedback.

---

> ### Author Response · Authors · 2025-11-27
>
> Dear Reviewer 5Gm1,
>
> We would like to respectfully follow up regarding our rebuttal. We understand the significant workload during the review period and appreciate the time you devote to evaluating submissions.
>
> If you have an opportunity, we would be grateful for any further comments or clarifications. Your feedback is highly valued.
>
> Thank you for your time and consideration.

---

### Meta-Review · Area_Chair_sRrf · 2026-01-08

**Summary:**

The main concerns are: 1) Major Conceptual Mismatch: The paper's single-policy formulation conflicts with its multi-policy algorithm, creating foundational confusion. 2) Unproven Novelty: Failure to properly position against highly relevant prior work like FedRLHF critically undermines claims of innovation. 3) Insufficient Justification for Core Mechanism: No theoretical or empirical analysis proves that aggregating policies is better than aggregating model weights. 4) Unsubstantiated Personalization: Experiments only show data heterogeneity, not preference heterogeneity with distinct reward models, leaving the key claim unverified.

**Reviewer Concerns:**

The rebuttal partially addressed peripheral concerns, clarifying that LoRA weights are shared (not raw policies) and explaining the role of classical RL experiments. However, the most critical concerns remain outstanding. These are: the fundamental theory-experiment mismatch in the core formulation; the lack of a proper comparison to FedRLHF; the absence of a rigorous justification for policy-space aggregation over weight aggregation; and the failure to demonstrate true preference-based personalization. The rebuttal's clarifications do not resolve these foundational issues.

**Reviewer Scores:**

I think most reviewers would likely maintain their reject scores after discussion. While some minor clarifications were offered, the discussion would likely amplify the consensus that the paper's core contribution is inadequately justified against the state-of-the-art, and lacks the necessary experiments to prove its primary claim.

---

### Decision · Program_Chairs · 2026-01-26

Reject